# McaA and McaB control the dynamic positioning of a bacterial magnetic organelle

Juan Wan [1], Caroline L. Monteil[2,6], Azuma Taoka[3,6], Gabriel Ernie[1], Kieop Park[1,4], Matthieu Amor [1,2], Elias Taylor-Cornejo [1,5], Christopher T. Lefevre[2] & Arash Komeili [1] ✉

Magnetotactic bacteria are a diverse group of microorganisms that use intracellular chains of ferrimagnetic nanocrystals, produced within magnetosome organelles, to align and navigate along the geomagnetic field. Several conserved genes for magnetosome formation have been described, but the mechanisms leading to distinct species-specific magnetosome chain configurations remain unclear. Here, we show that the fragmented nature of magnetosome chains in *Magnetospirillum magneticum* AMB-1 is controlled by genes *mcaA* and *mcaB*. McaA recognizes the positive curvature of the inner cell membrane, while McaB localizes to magnetosomes. Along with the MamK actin-like cytoskeleton, McaA and McaB create space for addition of new magnetosomes in between pre-existing magnetosomes. Phylogenetic analyses suggest that McaA and McaB homologs are widespread among magnetotactic bacteria and may represent an ancient strategy for magnetosome positioning.

Cellular compartmentalization results in the formation of different organelles, which need to be positioned correctly to fulfill their specific functions and ensure proper inheritance throughout cell division[1]. Organelle positioning in eukaryotic cells mainly relies on cytoskeletal and motor proteins[1]. Many bacteria also produce organelles[2–7], and actively regulate their placement in the cell. For example, the protein-bounded carbon-fixation organelle, the carboxysome, uses the nucleoid as a scaffold with helper proteins that ensure equal distribution in the cell and proper segregation into daughter cells[6]. Similarly, it has been proposed that the carbon storage poly-hydroxybutyrate granules associate with nucleoids to mediate segregation during cell division[7,8]. A widely studied example of bacterial lipid-bounded organelles is the magnetosome compartment of magnetotactic bacteria (MTB). Magnetosomes mineralize ferrimagnetic nanoparticles composed of magnetite ($Fe_3O_4$) and/or greigite ($Fe_3S_4$)[2,9], which are used as a compass needle for navigation along the geomagnetic field. Magnetic navigation is a common behavior in diverse organisms, including bacteria, insects, fish, birds, and mammals[10,11]. MTB are the simplest and most ancient organism capable

of magnetic navigation[12] and fossilized magnetosome chains have been used as robust biosignatures[13,14]. Thus, magnetosome production in MTB is an ideal model system for studying mechanisms of organelle positioning, understanding the evolution of magnetic navigation, and connecting the magnetofossil record to the history of life on Earth.

To function as an efficient compass needle, individual magnetosomes need to be arranged into a chain. Various and complex magnetosome chains (single- or multi-stranded, continuous, or fragmented) are found in diverse MTB groups[15–17]. The mechanisms leading to distinct chain configurations remain unknown, but may reflect strategies for adaptations to specific biotopes[13]. The most widely studied model MTB strains *Magnetospirillum magneticum* AMB-1 (AMB-1) and *Magnetospirillum gryphiswaldense* MSR-1 (MSR-1) are closely related *Alphaproteobacteria* species sharing 96% identity in their 16S rRNA gene sequences[18]. However, their magnetosome chain organization strategies are distinct. In AMB-1, magnetosomes containing magnetic crystals and empty magnetosomes (EMs) are interspersed to form a chain that is fragmented in appearance, extends from pole to pole in the cell, and remains stationary during the entire

[1]Department of Plant and Microbial Biology, University of California, Berkeley, CA 94720, USA. [2]Aix-Marseille Université, CEA, CNRS, Institute of Biosciences and Biotechnologies of Aix-Marseille, 13108 Saint-Paul-lez-Durance, France. [3]Institute of Science and Engineering, Kanazawa University, Kakuma-machi, Kanazawa, Ishikawa 920-1192, Japan. [4]Department of Biology, Duke University, Box 90338, Durham, NC 27708, USA. [5]Department of Biology, Randolph-Macon College, Ashland, VA 23005, USA. [6]These authors contributed equally: Caroline L. Monteil, Azuma Taoka. ✉e-mail: komeili@berkeley.edu

cell cycle[19]. In contrast, in MSR-1, magnetic crystals are arranged as a continuous chain at the midcell and the divided daughter chains rapidly move from the new poles to the center of the daughter cells after cell division[20,21]. The actin-like protein MamK is conserved in all characterized MTB and forms a cytoskeleton that specifically regulates the stationary or moving behaviors of magnetosome chains in AMB-1 and MSR-1[19,20]. While the overall proteomes of AMB-1 and MSR-1 are on average 66% identical, MamK proteins from the two organisms are 90.8% identical at the amino acid level and $mamK_{AMB-1}$ complements the MSR-1 $\Delta mamK$ mutant[22]. However, the speed and spatial dynamics of MamK filaments are distinct in each organism[19,20]. The acidic protein MamJ is also a key regulator of chain organization. When $mamJ$ is deleted in MSR-1, magnetosomes collapse into aggregates in cell[23]. However, similar deletions in AMB-1 result in subtle defects with magnetosomes still organized as chains[24]. In addition, $mamY$ is critical for localizing the chain to the positive curvature of the cell in MSR-1[25] but does not have an impact on chain organization when deleted in AMB-1[26]. These observations suggest that unknown genetic elements may be needed for species-specific chain organization phenotypes.

The genes for magnetosome production and chain assembly in MTB, such as $mamK$, $mamJ$, and $mamY$, are arranged into magnetosome gene clusters (MGCs) that are often structured as magnetosome gene islands (MAI)[27]. Unlike MSR-1, AMB-1 contains an extra genomic cluster, termed the magnetotaxis islet (MIS), outside of the MAI region[28]. The MIS is ~28 kb long and contains seven magnetosome gene homologs, including a very divergent copy of $mamK$ and many genes of unknown function[28]. The MIS protein, MamK-like, partners with MamK in magnetosome chain formation, but does not contribute to the species-specific chain organization phenotypes mentioned above[29]. Whether the other MIS genes are functional and play roles in magnetosome biosynthesis is unclear, especially given the presence of multiple transpose genes and pseudogenes[28].

Here, we studied the MIS genomic region and identified two proteins (McaA and McaB) that mediate magnetosome chain assembly in AMB-1. McaA localizes to the positively curved cytoplasmic membrane as a dashed line even in the absence of magnetosomes whereas McaB associates with magnetosomes. Together, McaA and McaB direct the addition of new magnetosomes to multiple sites between pre-existing magnetosomes to form a fragmented crystal chain. They also influence the dynamics of MamK filaments to control magnetosome positioning during the entire cell cycle. The action of McaA and McaB is sufficient to explain all of the known differences in chain organization between AMB-1 and MSR-1. Broader phylogenetic analysis reveals that McaA and McaB are specific to AMB-1 with distant homologs in the vicinity of MGCs in other MTB. We hypothesize that the MIS is a remnant of an ancient duplication event that paved the way for an alternative chain segregation strategy in AMB-1. This mode of chain segregation may lower the energy requirements for separating magnetic particles at the division septum and eliminate the need for rapidly centering the chain after cell division.

## Results

### MIS genes control the location of and spacing between magnetosomes

To investigate its possible role in magnetosome formation and placement, we deleted the entirety of the MIS from the AMB-1 genome. The growth curves of wild-type (WT) and ΔMIS strains are similar (Supplementary Fig. 1a and Supplementary Note 1). The deletion's effect on magnetosome production was then assessed by measuring the coefficient of magnetism (Cmag) using a differential spectrophotometric assay that quantifies the ability of MTB to orientate in an external magnetic field[30]. Unexpectedly, the Cmag values of ΔMIS cultures are much higher than WT cultures (Fig. 1a), indicating that, as a population, ΔMIS cells better align with the applied external magnetic field. As expected, transmission electron microscopy (TEM)

images of WT AMB-1 cultures show that the magnetic crystals are organized into a chain with gaps from cell pole to pole (Fig. 1b and Supplementary Fig. 1b). In contrast, the crystals in the ΔMIS strain are organized into a continuous chain at the midcell (Fig. 1b and Supplementary Fig. 1b). Analysis of TEM images shows that the number and length of crystals are similar (Supplementary Fig. 1c, d), but the shape factor of crystals (width/length ratio) differs between WT (0.82) and ΔMIS (0.92) strains (Supplementary Fig. 1e, f). These data were collected from strains grown under microaerobic conditions. To ensure that the observed phenotypes were not generated by specific growth conditions, the experiment was repeated under anaerobic conditions and yielded similar results (Supplementary Fig. 2).

Magnetosome biogenesis in WT AMB-1 begins with the invagination of the bacterial inner membrane to form EMs, followed by the crystallization of ferrimagnetic minerals to form crystal-containing magnetosomes (CMs)[2,9]. To directly observe the organization of magnetosome membranes, we imaged WT and ΔMIS cells with whole-cell cryo-electron tomography (cryo-ET). Magnetosomes are arranged as a chain that is flanked by a network of short filaments in both strains. Specifically, magnetosomes are located to the positive inner curvature of the cell (displayed along the area that is bent inward toward the cytoplasmic membrane), MamK filaments run parallel to two to five individual magnetosomes along the chain and do not show an obvious spatial position pattern in both WT and ΔMIS strains (Fig. 1c, d and Supplementary Movies 1 and 2). Similar to WT[31], the magnetosome membranes of ΔMIS mutant are invaginations of the inner membrane (Fig. 1d, lower left corner). We then measured the diameter of magnetosome membranes and the length of crystals. The size distribution of EMs and CMs, as well as the linear relationship between the sizes of crystals and magnetosome membrane diameters, is similar between the WT and ΔMIS mutant (Supplementary Fig. 3). In addition, in both strains, the EMs are significantly smaller than the CMs (Supplementary Fig. 3a). One major phenotypic difference between the two strains is that in WT EMs are present at multiple sites between CMs in the magnetosome chain (Fig. 1c) whereas EMs only localize at both ends of the continuous chain in the ΔMIS strain (Fig. 1d). Analyzing the size and biomineralization status of magnetosome membranes relative to their subcellular position shows that the location of EMs is random along the magnetosome chain in WT but is at both ends of the chain in ΔMIS cells (Fig. 1e, f). These results together suggest that MIS genes control the location of magnetosomes but not the size of magnetosome membranes.

Based on the different sizes and locations of magnetosomes, we hypothesized that newly made magnetosomes are added at multiple internal sites of the chain in WT, but only added at the ends of the chain in ΔMIS. To test this hypothesis, we designed a pulse-chase experiment to label and follow a marker protein that incorporates into magnetosomes at the early steps of membrane invagination (Fig. 2a). We examined the magnetosome marker proteins MamI and MmsF[32–34]. Newly synthesized MmsF proteins incorporate into both the new and old magnetosomes (see more details in Supplementary Note 2 and Supplementary Fig. 4), indicating that it is not suitable for the pulse-chase experiment.

The transmembrane (TM) protein MamI is needed for EM invagination from the inner membrane of AMB-1[29,34]. We first checked the localization of MamI-GFP in WT and ΔMIS. Three-dimensional (3D) structured illumination fluorescent microscopy (SIM) imaging shows MamI-GFP localizes as a continuous line from cell pole to pole in WT AMB-1 (Fig. 2b), indicating localization to both the EMs and CMs. In ΔMIS, MamI-GFP only localizes in the middle of the cells in a pattern reminiscent of magnetosome organization as seen in cryo-ET images (Fig. 2b). In addition, MamI-GFP was also observed at the cytoplasmic membrane, outlining the cell with a weak fluorescence (Fig. 2b). We then performed pulse-chase experiments using MamI-Halo. The Halo-ligand JF549 was used as the pulse to mark old magnetosomes and the

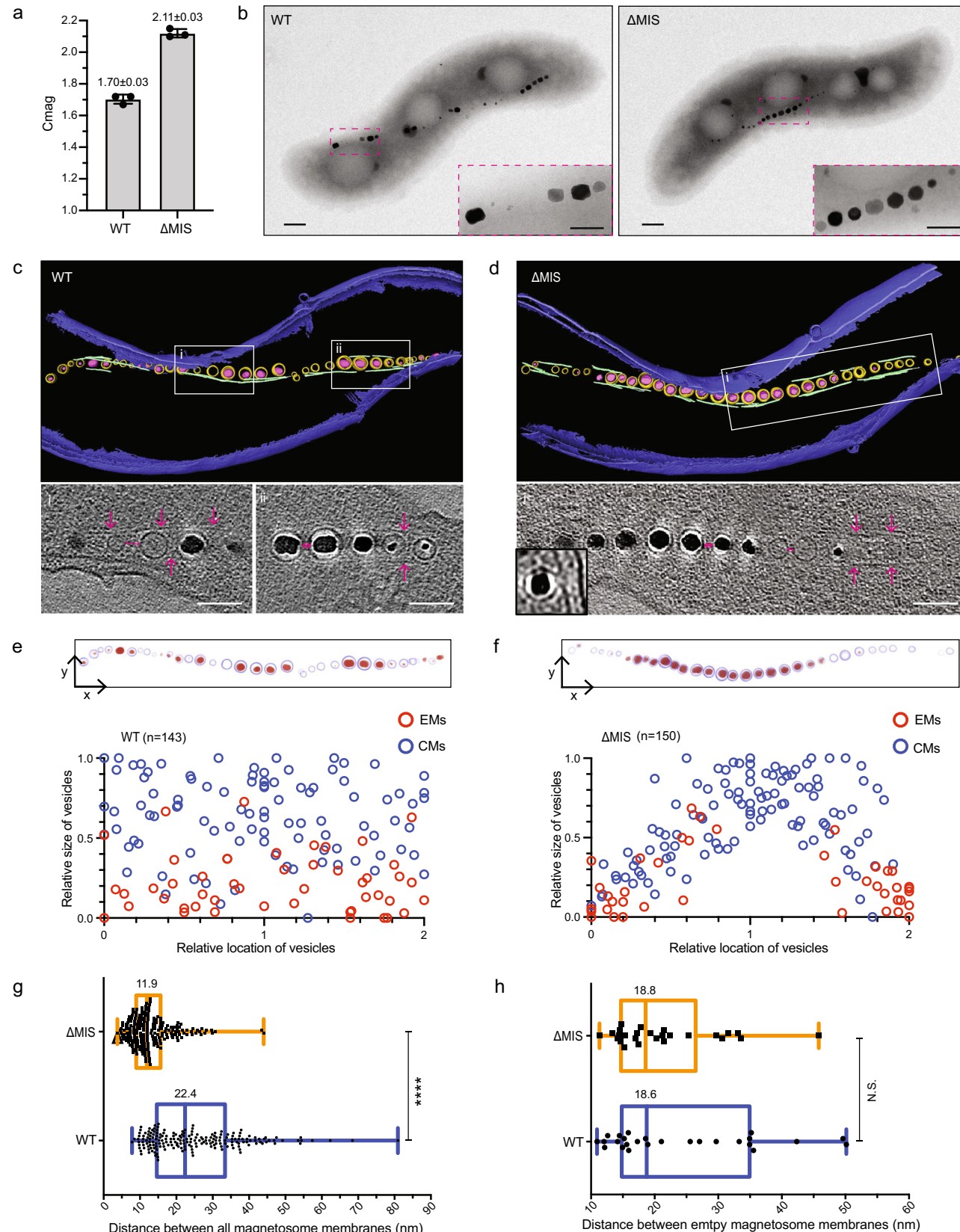

JF646 ligand was chased in to identify the newly made magnetosomes (Fig. 2a). JF646 signals do not colocalize with JF549 signals in WT AMB-1 (Fig. 2d and Supplementary Fig. 5a), indicating newly synthesized MamI proteins are only added to the newly made magnetosomes. Quantitative analysis shows very low colocalization coefficients of the

pulse and chase signals in WT and ΔMIS cells (Supplementary Table 1). As expected, JF549-marked old magnetosomes display gaps, which are filled with the JF646-marked newly made magnetosomes in WT AMB-1 (Fig. 2d and Supplementary Fig. 5a). Conversely, the JF549-marked old magnetosomes still mainly show a continuous chain at the midcell of

**Fig. 1 | MIS genes contribute to the magnetosome chain assembly. a** Magnetic response (Cmag) of WT and ΔMIS cultures grown under microaerobic conditions. Each measurement represents the average and standard deviation from three independent growth cultures. **b** TEM micrographs of WT and ΔMIS cells. Scale bars = 0.2 μm. Insets: magnification of the magnetic crystals in magenta rectangles. Insets scale bars = 100 nm. **c, d** Segmented 3D models (upper panels) and selected area of tomographic slices (lower panels, Box i and ii) showing phenotypes of WT and ΔMIS strains. Magenta arrows point to the MamK filaments on the tomographic slices. The outer and inner cell membranes are depicted in dark blue, magnetosome membranes in yellow, magnetic particles in magenta, and magnetosome-associated filaments in green. Scale bars = 100 nm. Full tomograms are shown in Supplementary Movies 1 and 2. **e, f** Relative size and location of magnetosome vesicles in WT and ΔMIS cells, respectively (lower panels). EMs empty magnetosomes, CMs crystal-containing magnetosomes. Upper panels are 2D projections of magnetosomes from the 3D models in **c** and **d**. The magnetosome membranes are shown in light blue and magnetic particles are shown in red. **g, h** Edge-to-edge distance (the magenta two-end arrows on the tomographic slices of **c** and **d**) between all of the magnetosomes (**g**) and the EMs (**h**) that were measured from neighboring magnetosome membranes in WT (blue) and ΔMIS (orange) strains. Values represent the median. $n = 199$ (WT) and 206 (ΔMIS) in **g**, $n = 26$ for both WT and ΔMIS in **h**. Box plots indicate median (middle line), 25th, 75th percentile (box), and min/max (whiskers). $P$ values were calculated by the two-sided Mann–Whitney $U$ test. No statistically significant difference ($P > 0.05$, N.S.), significant difference (****$P < 10^{-4}$). The source data of **a, e, f, g**, and **h** are provided as a Source Data file.

ΔMIS, and the JF646-marked newly made magnetosomes localize at both ends of the chain (Fig. 2d and Supplementary Fig. 5a). Together, these results confirm our hypothesis that the varying chain phenotypes between WT and ΔMIS strains are in part due to changes in the location where new magnetosomes are added.

In addition, we found that the length ratio between the MamI-GFP marked magnetosome chain and the cell body is significantly larger in WT than in ΔMIS (Fig. 2c). As mentioned above, the number of crystals in WT and ΔMIS cells is similar, indicating the distance between the magnetosomes might be different in these two strains. We therefore measured the edge-to-edge magnetosomes distance (magenta two-end arrows in Fig. 1c, d) and found that the distance between all magnetosomes in WT is about twice as long as in the ΔMIS strain (Fig. 1g), while the distance between EMs in these two strains is similar (Fig. 1h), indicating the difference is mainly due to the distance between CMs.

To summarize, MIS genes control the shape of crystals, the distance between CMs, and the location for the addition of newly made magnetosomes leading to the characteristic pattern of chain organization in AMB-1.

## Comprehensive dissection of the MIS chain organization factors

To identify the key genes that control magnetosome positioning, we conducted conventional recombination mutagenesis to create unmarked deletions of selected segments in the MIS (Fig. 3a). We first deleted large domains (LD1 and LD2) to narrow down the region of interest to LD1 (Fig. 3b, c and Supplementary Note 3). We then generated small islet region deletions (ΔiR1, ΔiR2, ΔiR3, and ΔiR4) of LD1 to pinpoint specific genes involved in chain organization. Genes in the iR2 region control magnetosome positioning, while those in iR3 contribute to crystal shape control (Fig. 3b, d and Supplementary Note 3). iR2 contains a small putative operon with two hypothetical genes (*amb_RS23835* and *amb_RS24750*), which we have named magnetosome chain assembly genes A and B (*mcaA* and *mcaB*) (Fig. 3a). The reference genome in NCBI shows the iR2 region includes a third transposase gene (*amb_RS23840*) (Fig. 3a), which does not exist in our lab strain. We then deleted these two genes individually. Both Δ*mcaA* and Δ*mcaB* strains have dramatically higher Cmag compared to WT and are similar to ΔMIS (Fig. 3b). Similar to the ΔMIS mutant, Δ*mcaA* and Δ*mcaB* strains contain continuous crystal chains in the midcell region when viewed by TEM (Fig. 3d), indicating that both play essential roles in magnetosome positioning.

## McaA localizes to the positively curved cytoplasmic membrane as a dashed line

We interrogated the localization of McaA and McaB in order to understand their role in controlling magnetosome positioning. McaA is predicted to contain a signal peptide, followed by a periplasmic von Willebrand factor type A (VWA) domain, a TM domain, and a cytoplasmic C-terminus (Fig. 4a and Supplementary Note 4). However, in some bioinformatic predictions, the signal peptide region is predicted to be a TM domain with the N-terminus facing the cytoplasm (Supplementary Table 2). Using GFP fusions to either end of the protein, we predict that the C-terminus of McaA faces the cytoplasm (See more details in Supplementary Note 4). 3D-SIM images show that, when expressed in WT AMB-1, McaA-GFP localizes in a dashed-line pattern distributed along the positive inner curvature of the cell (Fig. 4b–d). Specifically, McaA is close to the regions of the cell envelope that are bent inward toward the cytoplasm, forming a line covering the shortest distance from cell pole to pole (Fig. 4c and Supplementary Movie 3). Since magnetosomes are also located to the positive inner curvature of AMB-1 cell[25], we wondered if McaA is associated with magnetosomes. To test this, we expressed McaA-GFP in different genetic backgrounds and growth conditions, including ΔMIS and ΔiR2 (also called Δ*mcaAB*) mutants which contain continuous crystal chains, WT under iron starvation where only EMs are present (Fig. 4e, f), and the ΔMAIΔMIS mutant that is incapable of magnetosome production (Fig. 4g–i and Supplementary Movie 4). McaA-GFP localizes as a dashed line in all of the above strains and conditions (Fig. 4b–i), indicating that the association of McaA with the positive curvature of cytoplasmic membrane and its dashed-line localization are independent of magnetosome membrane formation, magnetite production, and chain organization.

Using a series of truncations, we found that the N-terminus of McaA (including the predicted signal peptide and the VWA domain) is essential for its localization and magnetosome positioning (Fig. 4j, k and Supplementary Note 5). In contrast, the C-terminus conserved region (aa 530-665) is not essential for McaA localization but important for magnetosome positioning (Fig. 4j, k and Supplementary Note 5). The VWA domain is commonly involved in protein–protein interactions, largely via a noncontiguous metal ion-binding motif (DXSXS) called metal ion-dependent adhesion site (MIDAS)[35,36]. McaA VWA domain contains an intact MIDAS motif (Fig. 4a). We investigated whether this MIDAS motif plays an important role using site-directed mutagenesis. McaA^MIDAS mutant cannot complement Δ*mcaA* (Fig. 4j and Supplementary Fig. 8) and is evenly distributed within the cytoplasmic membrane (Fig. 4k), highlighting the important role of the MIDAS motif and divalent cations in the localization and function of McaA.

## McaB associates with crystal-containing magnetosomes

McaB is predicted to contain one TM domain that is close to the N-terminus, which is mostly facing the periplasm (Supplementary Fig. 9a). We confirmed the cytoplasmic location of C-terminus McaB through the fluorescent signal of McaB-GFP (Fig. 5a and Supplementary Note 4). 3D-SIM images show that McaB-GFP forms a dotted line from cell pole to pole along the positive inner curvature of WT AMB-1 cells, whereas it exhibits a continuous line in the middle of the ΔMIS and Δ*mcaAB* cells (Fig. 5a and Supplementary Fig. 9b), indicating McaB might be associated with magnetosomes. Interestingly, McaB-GFP is not present at the magnetosome chain when WT cells are grown under low iron conditions that prevent magnetite production (Fig. 5a and Supplementary Fig. 9b), indicating

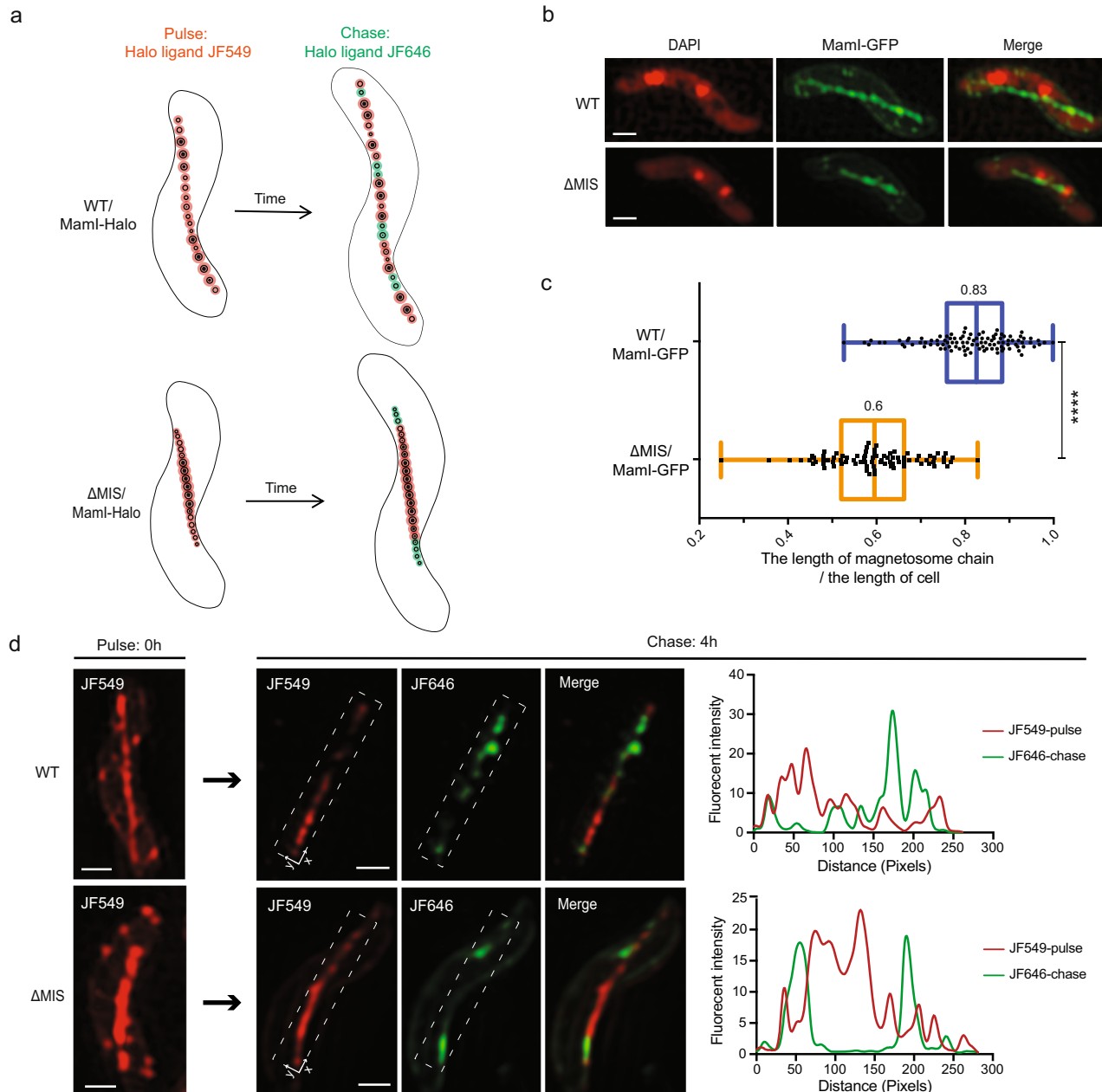

**Fig. 2 | Pulse-chase experiments that characterize the addition of newly formed magnetosomes to the chain. a** Model of the pulse-chase experiment shows how the newly formed EMs are added to the magnetosome chains in WT and ΔMIS cells. **b** Representative maximum-intensity projection of 3D-SIM micrographs (generated from 3D z-stacks) of WT and ΔMIS cells expressing MamI-GFP under standard growth conditions. MamI-GFP proteins are located in the magnetosome chain and cytoplasmic membrane. Here and below: 4',6-Diamidino-2-phenylindole (DAPI) is a fluorescent dye that binds to DNA, and used as an indicator of AMB-1 cell contour. The DAPI staining is shown in false-color red, and MamI-GFP is shown in green. **c** Quantification of the length of magnetosome chain versus the length of cell in WT

(blue) and ΔMIS (orange) strains. Values represent the median. $n = 103$ for WT, $n = 82$ for ΔMIS. Box plots indicate median (middle line), 25th, 75th percentile (box), and min/max (whiskers). $P$ value was calculated by the two-sided Mann−Whitney $U$ test. Significant difference (****$P < 10^{-4}$). The source data are provided as a Source Data file. **d** Representative maximum-intensity projection of 3D-SIM micrographs and fluorescent intensity maps (white dashed rectangular area) of the pulse-chase experiments with MamI-Halo fusion protein for analyzing the addition of newly formed magnetosomes in WT and ΔMIS cells. The JF549 staining is shown in red, and the JF646 staining is shown in green. Scale bars = 0.5 μm in **b** and **d**.

that it does not localize to the EMs. These results together show that McaB might be specifically associated with CMs (Supplementary Fig. 9c), resembling the localization pattern of the magnetosome protein Mms6[37]. The association of Mms6 with CMs has been addressed previously in AMB-1 through localization analysis under different growth conditions and correlative fluorescent and TEM microscopy analysis[37]. Thus, we co-expressed McaB-GFP and Mms6-Halo in the same AMB-1 cell, and the 3D-SIM images, as well as quantitative determination of colocalization coefficients

(Supplementary Table 1), show that the two proteins colocalize (Fig. 5b, c), further confirming the association of McaB with CMs.

We also examined the localization of McaA and McaB using cellular fractionation and immunoblotting analysis. McaA is mainly detected in the insoluble portion that includes the cytoplasmic membranes, while similar to Mms6, McaB is mainly detected in the magnetosome fraction (Fig. 5d). Thus, biochemical fractionation experiments confirm the association of McaA with cytoplasmic membrane and McaB with CMs.

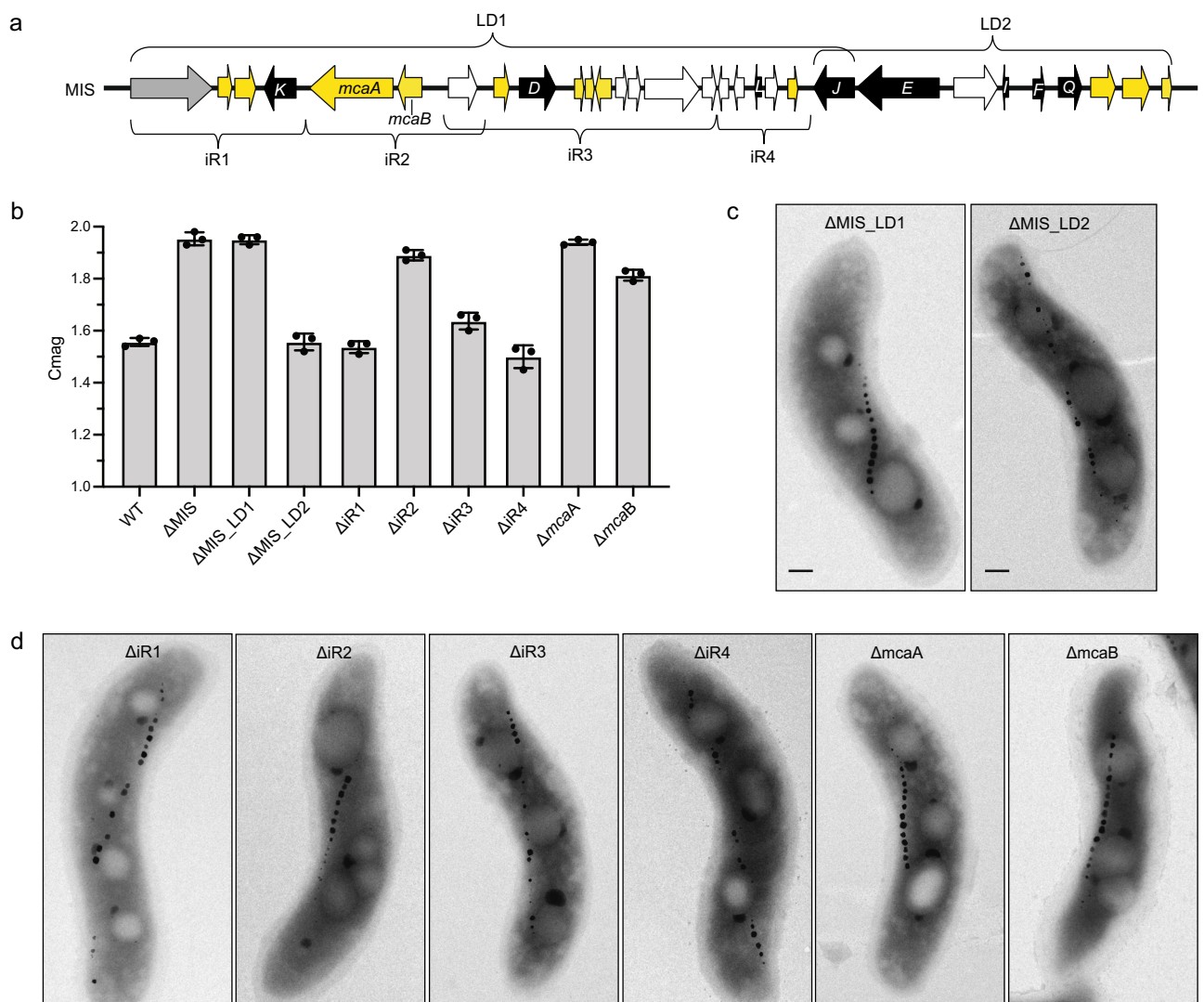

**Fig. 3 | Comprehensive genetic dissection of the MIS genes. a** Schematic depicting the MIS region of AMB-1, including the predicted magnetosome gene homologs (black), the gene of a phage-associated protein (gray), the genes of transposases (white), and the hypothetical genes (yellow). LD1 and LD2, large domains 1 and 2. iR1-iR4, small islet regions 1–4. *K, mamK-like*; *D, mamD-like*; *L, mamL-like*; *J, mamJ-like*; *E, mamE-like*; *I, mamI-like*; *F, mamF-like*; *Q, mamQ-like*.

*mcaA*, magnetosome chain assembly gene A. *mcaB*, magnetosome chain assembly gene B. **b** Cmag of WT and different mutants in the MIS region. Each measurement represents the average and standard deviation from three independent growth cultures. The source data are provided as a Source Data file. **c** TEM micrographs of ΔMIS_LD1 and ΔMIS_LD2 cells. Scale bars = 0.2 μm. **d** TEM micrographs of ΔiR1-ΔiR4, Δ*mcaA*, and Δ*mcaB* cells. Scale bars = 0.2 μm.

## McaA and McaB coordinate magnetosome positioning

To explore the relationship between McaA and McaB, we co-expressed McaA-Halo and MacB-GFP in WT AMB-1 cells (see more details in Supplementary Note 6). Interestingly, 3D-SIM images show that McaB localizes within the gaps of dashed McaA, indicating McaB-marked CMs are located in the gaps of dashed McaA (Fig. 5e, f and Supplementary Fig. 10d). Accordingly, quantitative analysis shows low colocalization coefficients for McaA and McaB signals in WT AMB-1 cells (Supplementary Table 1). Bacterial adenylate cyclase two-hybrid (BACTH) assays did not show any positive interactions between McaA and McaB (Table 1 and Supplementary Fig. 11a), indicating that either the fusion proteins do not interact strongly, are nonfunctional in the context of BACTH, or unknown intermediate proteins are needed to facilitate their interactions.

We next investigated whether McaA and McaB directly affect the positioning of EMs when CMs are not produced. We grew WT, ΔMIS, and Δ*mcaAB* strains expressing MamI-GFP under iron starvation conditions. 3D-SIM images show that MamI-marked EMs are located continuously in the midcell of all three strains (Fig. 6a), indicating that

McaAB do not participate in the chain organization under low iron conditions. We then examined the dynamics of magnetosome chain organization as WT and cells missing *mcaAB* transitioned from low to high iron conditions to trigger magnetite production in EMs (Fig. 6b). We performed pulse-chase experiments using MamI-Halo with cells growing from iron starvation (pulse with JF549) to standard iron growth conditions (chase with JF646). As expected, the pulse experiments show a continuous chain of EMs in the middle of all WT and *mcaAB* deficient cells under iron starvation conditions (Fig. 6c and Supplementary Fig. 5b). After iron addition to the growth medium, we observed the formation of gaps between JF549-marked old magnetosomes in WT, but not in ΔMIS or Δ*mcaAB* cells (Fig. 6c and Supplementary Fig. 5b). In addition, the JF646-marked newly made EMs filled the gaps between older magnetosomes in WT but were only added at both ends of the chain in ΔMIS and Δ*mcaAB* (Fig. 6c and Supplementary Fig. 5b). Accordingly, quantitative analysis shows a low colocalization coefficient of the pulse and chase signals in WT, ΔMIS, and Δ*mcaAB* cells (Supplementary Table 1). Together, these results support the hypothesis that McaA serves as a landmark on the positively curved

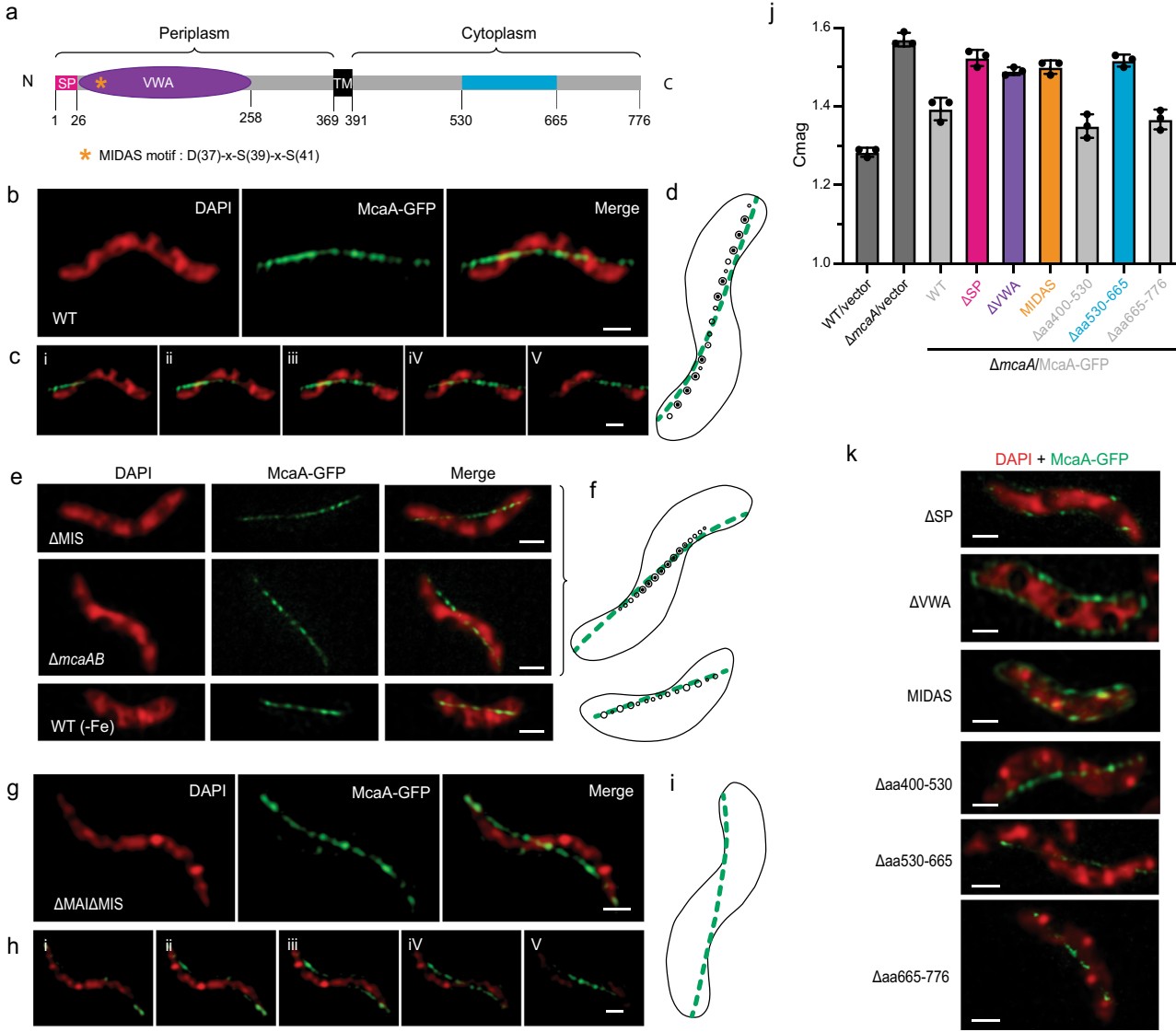

**Fig. 4 | Localization of McaA. a** Predicted secondary structure and topology of McaA. The whole protein is shown in a light gray line. The conserved C-terminus region is highlighted in blue. SP signal peptide (magenta line), VWA von Willebrand factor type A domain (purple oval), TM transmembrane domain (black rectangle), MIDAS metal ion-dependent adhesion site (orange star). **b** Representative maximum-intensity projection of 3D-SIM micrographs shows a WT AMB-1 cell expressing McaA-GFP. **c** Consecutive z-slices with a distant spacing of 100 nm from the merged channel of **b**. **d** A model of magnetosome production and McaA (green) localization in a WT AMB-1 cell. **e** Representative maximum-intensity projection of 3D-SIM micrographs shows cells expressing McaA-GFP in different genetic backgrounds or growth conditions. **f** Models of magnetosome production and McaA (green) localization from **e**. **g** Representative maximum-intensity projection of 3D-SIM micrographs shows a ΔMAIΔMIS cell expressing McaA-GFP. **h** Consecutive z-slices with a distant spacing of 100 nm from the merged channel of **g**. **i** A model of magnetosome production and McaA (green) localization in a ΔMAIΔMIS cell. **j** Cmag of WT and mutated McaA-GFP expressed in Δ*mcaA* cells. The colors of the mutated regions correspond to the colored regions in **a**. Each measurement represents the average and standard deviation from three independent growth cultures. The source data are provided as a Source Data file. **k** Representative maximum-intensity projection of 3D-SIM micrographs of mutated McaA-GFP expressed in Δ*mcaA* cells. The DAPI staining is shown in false-color red and the GFP fusion proteins are shown in green. Scale bars = 0.5 μm.

inner membrane and coordinates with McaB to control the location and spacing between CMs, allowing the addition of newly made EMs to multiple sites between pre-existing magnetosomes in the chain of WT AMB-1, which forms the fragmented crystal chain.

## McaA contributes to the differences of *mamJ* and *mamY* deletions between AMB-1 and MSR-1

As mentioned above, the phenotypes of *mamJ* and *mamY* deletion mutants in AMB-1 and MSR-1 are distinct. MamJ is proposed as a linker to attach MamK filaments to magnetosomes and its deletion in MSR-1 causes the magnetosome chain to collapse and form an aggregate[23]. In contrast, the deletion of *mamJ* and its homolog *limJ* in AMB-1 still shows a magnetosome chain with some minor structural defects[24].

However, deletion of the entire MIS in a Δ*mamJ*Δ*limJ* strain causes a dramatic chain collapse phenotype resembling those of MSR-1 Δ*mamJ* mutant[13,23] (Fig. 7a, c). MIS contains a second *mamJ* homolog called *mamJ-like* and our deletion analysis shows that it does not contribute to chain maintenance (Fig. 3a and Supplementary Note 7). To figure out the specific genes, we generated large domain and small region deletion mutants of the MIS in a Δ*mamJ*Δ*limJ* background. Cmag and TEM images show that *mcaA* is the specific gene that prevents magnetosome aggregation in the Δ*mamJ*Δ*limJ* strain (Fig. 7a, c and Supplementary Fig. 13).

MamY is a membrane protein that directs magnetosomes to the positively curved inner membrane in MSR-1, thus aligning the magnetosome chain to the motility axis within a helical cell[25]. When *mamY* is

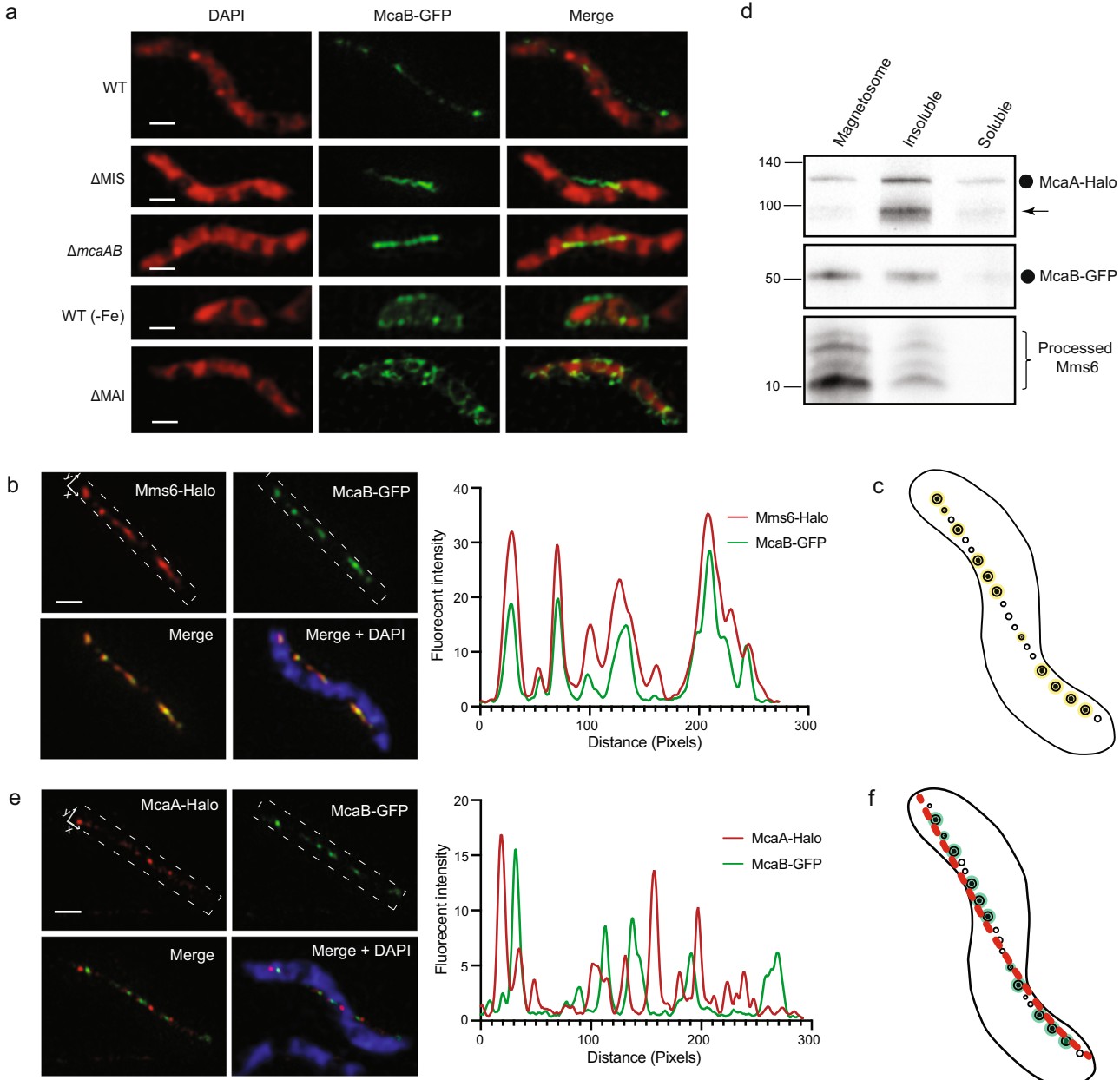

**Fig. 5 | Localization of McaB. a** Representative maximum-intensity projection of 3D-SIM micrographs shows the localization of McaB-GFP in WT and different genetic backgrounds. The DAPI staining is shown in false-color red and the GFP fusion proteins are shown in green. **b** Representative maximum-intensity projection of 3D-SIM micrographs and fluorescent intensity map (white dashed rectangular area) show the localization of Mms6-Halo and McaB-GFP in WT. **c** A model of the colocalization (yellow) of Mms6 and McaB at CMs from **b. d** Immunoblotting shows that McaA and McaB are enriched in different cellular fractionations. McaA-Halo was detected with anti-Halo antibodies, McaB-GFP was detected with anti-GFP antibodies, and Mms6 was detected with anti-Mms6 antibodies. Full-length McaA-Halo (~118 kDa) and McaB-GFP (~52 kDa) proteins are marked with a circle. The unknown McaA-Halo-related bands are marked with an arrow. The four proteolytically processed Mms6 fragments are marked with a right brace. The source data are provided as a Source Data file. See more details and controls in Supplementary Fig. 10e–h. **e** Representative maximum-intensity projection of 3D-SIM micrographs and fluorescent intensity map (white dashed rectangular area) show the localization of McaA-Halo and McaB-GFP in WT. **f** A model of the association of McaA (red) and McaB (green) with magnetosomes from **e. In b** and **e**, the DAPI staining is shown in blue, the JF549-stained Halo proteins are shown in red, and the GFP fusion proteins are shown in green. Scale bars = 0.5 μm.

deleted in MSR-1, the magnetosome chain is no longer restricted to the positively curved regions of the membrane and can also be found at the negatively curved membrane leading to a much lower Cmag compared to WT[25]. Surprisingly, when *mamY* is deleted in AMB-1, the Cmag is similar to WT (Fig. 7b), and the magnetosome chain still localizes to the positively curved membrane (Fig. 7d), indicating there might be other proteins that are functionally redundant to MamY in AMB-1. A Δ*mamY*ΔMIS mutant of AMB-1 has a much lower Cmag than ΔMIS and produces magnetosome chains that localize to both positively and negatively curved cell membranes (Fig. 7b, d). Further deletion mutagenesis shows that McaA helps magnetosomes localize to the positive-curved membrane when MamY is lost in AMB-1 (Fig. 7b, d).

Despite their genetic interactions, BACTH analysis does not show any direct interactions between MamJ and McaA, -B or between MamY and McaA, -B (Table 1 and Supplementary Fig. 11b–e). Nevertheless, our results indicate that the activity of McaA accounts for the distinct phenotypes of *mamJ* or *mamY* deletion mutants between AMB-1 and MSR-1.

**Table 1 | The interaction results of BACTH**

| | Zip-T25 | T25-McaA | McaA-T25 | T25-McaB | McaB-T25 | T25-MamK | MamK-T25 | T25-MamJ | T25-MamY | MamY-T25 |
|---|---|---|---|---|---|---|---|---|---|---|
| Zip-T18 | + | NA | NA | NA | NA | NA | NA | NA | NA | NA |
| T18-McaA | NA | – | – | – | – | – | – | – | – | – |
| McaA-T18 | NA | – | – | – | – | – | – | – | – | – |
| T18-McaB | NA | – | – | – | – | – | – | – | – | – |
| McaB-T18 | NA | – | – | – | – | – | – | – | – | – |
| T18-MamK | NA | – | – | – | – | + | + | NA | NA | NA |
| MamK-T18 | NA | – | – | – | – | + | + | NA | NA | NA |
| MamJ-T18 | NA | – | – | – | – | NA | NA | – | NA | NA |
| T18-MamY | NA | – | – | – | – | NA | NA | NA | – | – |
| MamY-T18 | NA | – | – | – | – | NA | NA | NA | – | + |

*NA* no analysis was performed. – negative interaction (white colonies), + positive interaction (blue colonies).

## McaAB controls magnetosome dynamic positioning by influencing MamK filaments

In addition to the appearance of the chain, the dynamic movements and positioning of the chain differ between AMB-1 and MSR-1[19,20]. We reasoned that the McaAB system may contribute to the different dynamic chain positioning in these two strains. Using highly inclined and laminated optical sheet (HILO) microscopy, we performed live-cell imaging analysis to follow the dynamics of Mms6-GFP labeled magnetosomes during cell division in WT and McaAB deficient cells.

In WT AMB-1, magnetosomes are in static, spotty positions during cell division as indicated by the parallel lines in the kymographs of GFP fluorescence (Fig. 8a and Supplementary Movie 5), whereas every ΔMIS and Δ*mcaA* cell shows dynamic magnetosome chain segregation after cell division (Fig. 8b, c and Supplementary Movies 6 and 7). In ΔMIS and Δ*mcaA*, magnetosomes are positioned at the midcell until the cell divides. After cytokinesis, magnetosomes are moved synchronously toward the centers of both daughter cells. Magnetosome migration to the middle of the daughter cells was completed within about 1 h after cell division. Magnetosome chain displacement velocity in Δ*mcaA* cells was about 20 nm min⁻¹, in line with what has been reported previously for MSR-1[20]. Supplementary Movies 9 and 10 are long time-lapse videos of the ΔMIS and Δ*mcaA* cells, in which the magnetosomes are stably positioned in the middle of the daughter cells during the entire cell cycle after the migration of the magnetosomes. In contrast, in Δ*mcaB* cells, magnetosome displacements are incomplete (Fig. 8d and Supplementary Movies 8 and 11 (short and long time-lapse)). Magnetosomes do not move synchronously toward the middle of the cell and are randomly positioned in the cell. In other words, these results show that McaB does not impact daughter chain positioning in Δ*mcaA* strain, while McaA impedes daughter chain positioning in Δ*mcaB* strain, indicating McaA might have extra functions in magnetosome chain positioning during cell division.

It has been shown that the dynamics of MamK filaments are essential for magnetosome chain positioning in AMB-1 and MSR-1[19,20]. To test whether the McaAB system influences the dynamics of MamK filaments, we performed fluorescence recovery after photobleaching (FRAP) assays on MamK-GFP filaments in WT and McaAB deficient cells. GFP-tagged MamK filaments localize as even thin lines from cell pole to pole in both WT and McaAB deficient AMB-1 strains. During FRAP experiments, sections of GFP-tagged MamK filaments are irreversibly photobleached and the recovery of fluorescence in the bleached area is tracked over time (Fig. 8e). The half-life ($t_{1/2}$) of recovery represents the time point at which 50% of the fluorescence intensity returns to the bleached region relative to the whole filament at that same time point. The bleached area does not move in WT (Fig. 8e and Supplementary Fig. 14a), but it moves in a fraction of ΔMIS, Δ*mcaA*, and Δ*mcaB* cells (Fig. 8g, Supplementary Fig. 14b, and

Supplementary Table 3). The $t_{1/2}$ fluorescence recovery is similar in WT and McaAB deficient cells containing an immotile bleached spot (Fig. 8f and Supplementary Fig. 14a). For the McaAB deficient cells with a moving bleached spot, the $t_{1/2}$ fluorescence recovery of the original bleached area is similar but much faster than in the cells containing an immotile bleached spot (Fig. 8h), indicating a similar moving speed of the bleached area. BACTH analysis did show MamK self-interactions but did not show any interactions between McaA, -B, and MamK (Table 1 and Supplementary Fig. 11f, g). Together, these results indicate that McaA, -B influence the dynamics of MamK filaments which, in turn, leads to the AMB-1-specific pattern of magnetosome chain organization.

## *mcaA* and *mcaB* genes are specific to MTB

To understand the evolutionary origins of the *mcaAB* system, we searched for homologs of these two genes in diverse species of MTB. Distant homologs were found in 38 MTB species. All of them belong to MTB strains either with characterized magnetosome chain phenotypes (Fig. 9a, b) or metagenomes obtained from a magnetic enrichment (Supplementary Fig. 15a, b), with the majority affiliated to the *Rhodospirillaceae* family in the *Alphaproteobacteria* class. Based on published reports, most studied MTB contain continuous crystal chains (Supplementary Dataset 1). However, some species show fragmented crystal chains, including the two *Alphaproteobacteria* species (*Ca.* Terasakiella magnetica PR-1 and *Terasakiella sp.* SH-1) that contain distant *mcaAB* homologs and two *Deltaproteobacteria* species that do not contain *mcaAB* homologs (Fig. 9a, b and Supplementary Dataset 1), which suggests that different mechanisms may exist for crystal chain fragmentation.

Besides the *mcaAB* genes of the MIS, AMB-1 contains two additional homologs of *mcaA* and *mcaB* with a similar domain architecture (named *mcaA-like* and *mcaB-like* here) that are present close to the MAI. McaA-like (encoded by *amb0908/amb_RSO4660*) has ~45% amino acid sequence identity over ~66% McaA sequence length, while McaB-like (encoded by *amb0907/amb_RS24855*) has ~44% amino acid sequence identity over ~66% McaB sequence length. We deleted *mcaAB-like* genes in both WT and ΔMIS AMB-1 strains but did not observe any obvious defects or changes in magnetosome formation or chain organization (Supplementary Fig. 15c, d), indicating *mcaAB-like* genes are not functionally redundant with *mcaAB* genes in AMB-1. Based on comparative genomic analyses, *mcaA* and *mcaB* of AMB-1 form a distinct cluster, while *mcaA-like* and *mcaB-like* of AMB-1 cluster with the second group of homologs in the strains that do not show a fragmented crystal chain phenotype (Fig. 9a, b, clade in light blue), indicating those homologs might have functions more similar to *mcaAB-like* genes in AMB-1.

Molecular phylogenetics indicates that the last common ancestor to Mca proteins of AMB-1 and Mca-like proteins detected in

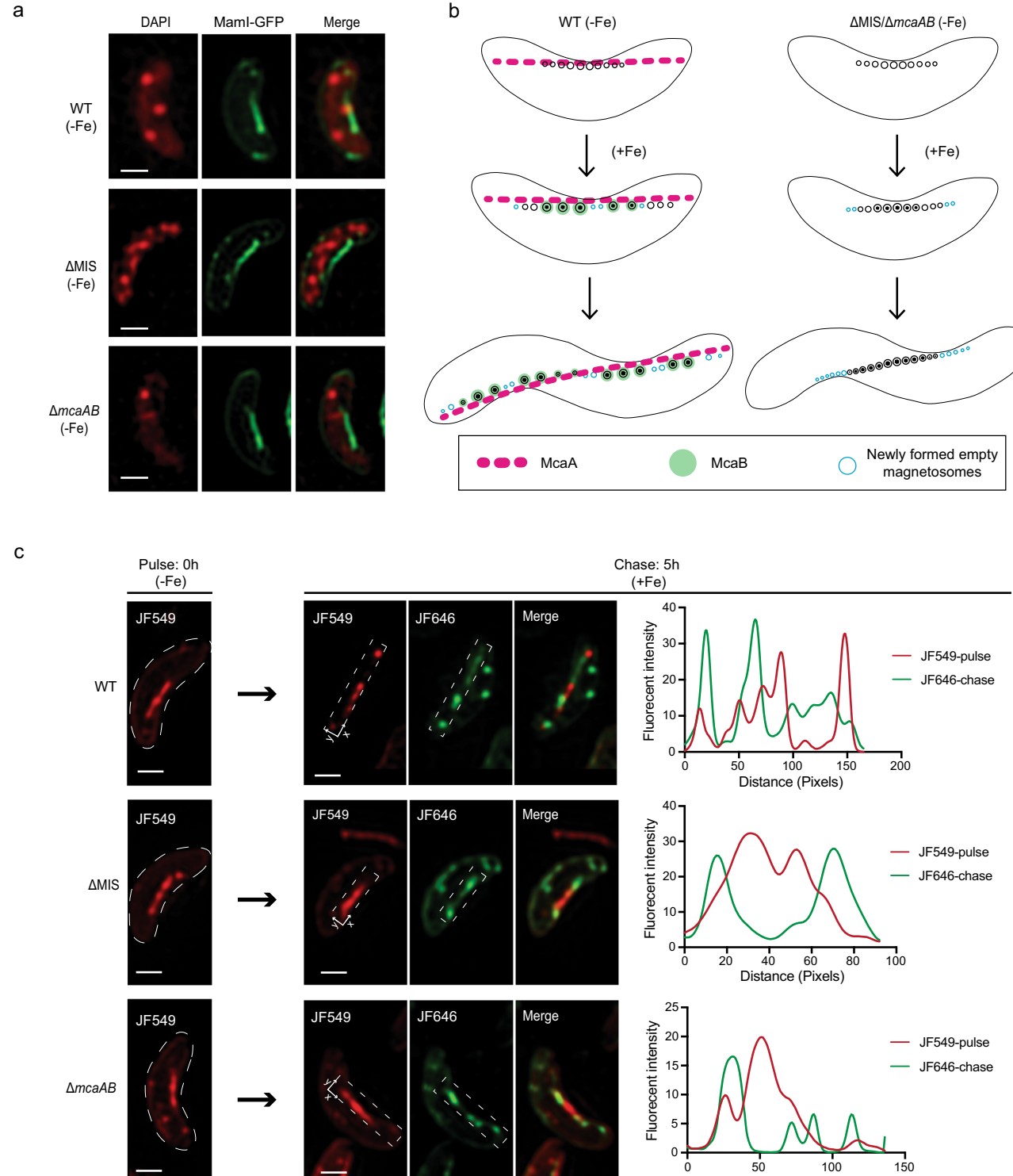

**Fig. 6 | Addition of newly formed magnetosomes to the chain from iron starvation to standard iron growth conditions. a** Representative maximum-intensity projection of 3D-SIM micrographs of WT, ΔMIS, and Δ*mcaAB* cells expressing MamI-GFP under iron starvation growth conditions. The DAPI staining is shown in false-color red, and MamI-GFP is shown in green. MamI-GFP proteins are located in the magnetosome chain and cytoplasmic membrane. **b** A model of how McaA and McaB coordinate to control the addition of newly formed magnetosomes in WT and ΔMIS/Δ*mcaAB* cells. A continuous chain of EMs is generated in the middle of both WT and McaAB deficient cells under iron starvation conditions. After the iron is added to the medium, magnetic crystals are mineralized in the existing EMs[33,59]. Once EMs become CMs, the McaB is recruited to CMs and helps locating CMs to the

gaps of dashed McaA, thereby increasing the distance between CMs, which allows the addition of newly formed EMs between CMs in WT AMB-1. In the absence of McaAB, CMs are closely located with each other, and the newly formed EMs have to be added at both ends of the chain. **c** Representative maximum-intensity projection of 3D-SIM micrographs and fluorescent intensity maps (white dashed rectangular area) of the pulse-chase experiments with MamI-Halo fusion protein for analyzing the addition of newly formed magnetosomes in different AMB-1 genetic backgrounds from iron starvation to standard iron growth conditions. The JF549 staining is shown in red, and the JF646 staining is shown in green. Scale bars = 0.5 μm in **a** and **c**.

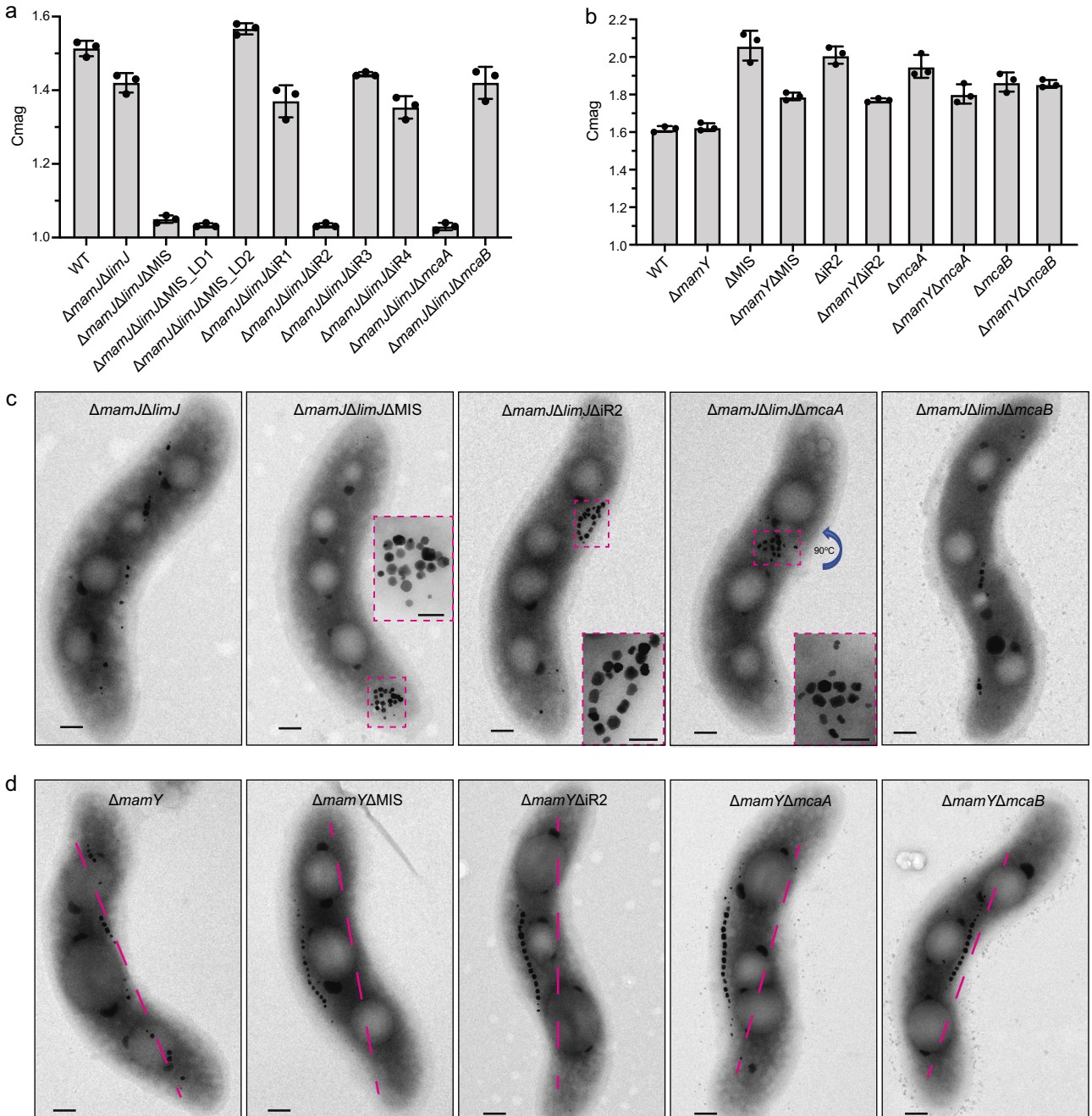

**Fig. 7 | McaA prevents magnetosome chain aggregation and directs the chain to the positively curved cytoplasmic membrane in AMB-1. a, b** Cmag of WT and different mutants in the Δ*mamJ*Δ*limJ* (**a**) and Δ*mamY* (**b**) backgrounds. Each measurement represents the average and standard deviation from three independent growth cultures. The source data are provided as a Source Data file. **c** TEM micrographs of Δ*mamJ*Δ*limJ*, Δ*mamJ*Δ*limJ*ΔMIS, Δ*mamJ*Δ*limJ*ΔiR2 (also called Δ*mamJ*Δ*limJ*Δ*macAB*), Δ*mamJ*Δ*limJ*Δ*mcaA*, and Δ*mamJ*Δ*limJ*Δ*mcaB* cells. Scale bars = 0.2 μm. Insets: magnification of the magnetic crystals in magenta rectangles. Insets scale bars = 100 nm. **d** TEM micrographs of Δ*mamY*, Δ*mamY*ΔMIS, Δ*mamY*-ΔiR2 (also called Δ*mamY*Δ*macAB*), Δ*mamY*Δ*mcaA*, and Δ*mamY*Δ*mcaB* cells. The magenta dashed lines indicate the positive inner curvature of AMB-1 cells where magnetosomes are normally located. Scale bars = 0.2 μm.

*Magnetospirillum* species and *Ca*. Magneticavibrio boulderlitore LM-1 emerged before the first freshwater magnetotactic *Rhodospirillaceae* (Fig. 9a, b). High protein sequence identity percentages of Mca-like proteins between *Magnetospirillum* strains (85% up to 100%) compared to the average amino acid ID % at the genome scale[38] indicates that the *mca-like* genes are under purifying selection. Together these results may indicate that McaAB-mediated crystal chain fragmentation could be a trace of an ancient chain organization strategy that is lost in most of the modern MTB species. However, the long external branches of McaA and McaB (Fig. 9a, b and Supplementary Fig. 15a, b) suggest a

recent acceleration of the evolution of Mca proteins that could be linked with their neofunctionalization in AMB-1.

## Discussion

In this study, we discovered a magnetosome chain organization strategy that explains phenotypic differences between closely related MTB. We demonstrated that the fragmented crystal chain organization in WT AMB-1 strain is not growth condition dependent but genetically controlled by *mcaA* and *mcaB*. In their absence, magnetite crystals form a continuous chain similar to the chain phenotype of MSR-1[25]. The

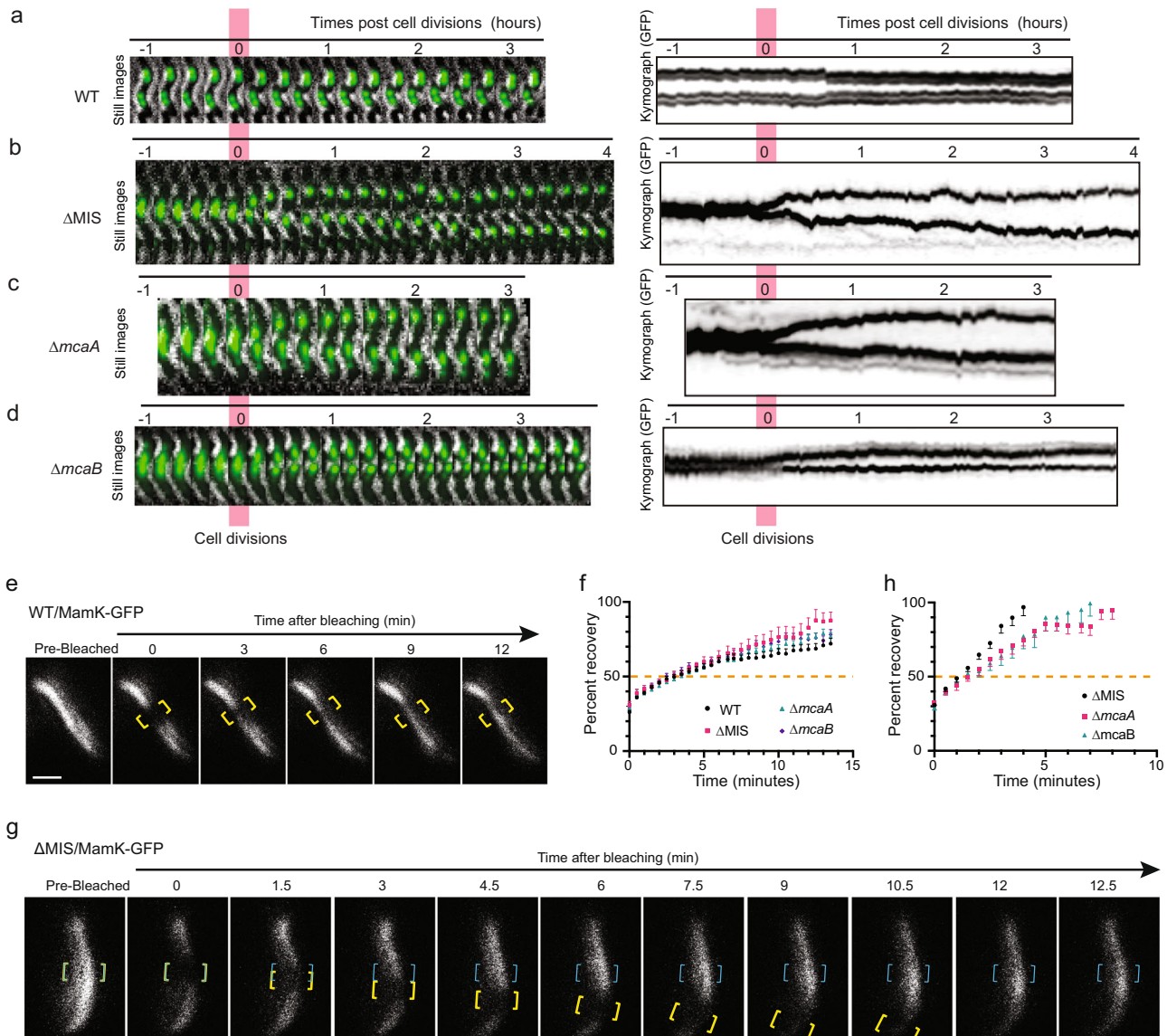

**Fig. 8 | Live-cell imaging shows the behavior of magnetosome chains and dynamics of MamK filaments. a–d** Effects of *mcaA* and *mcaB* deletions on magnetosome segregation. Live-cell time-lapse imaging of magnetosome segregation in WT (**a**), ΔMIS (**b**), Δ*mcaA* (**c**), and Δ*mcaB* (**d**) cells during cell division. Magnetosomes were labeled with Mms6-GFP. Left: GFP fluorescence and bright-field merged time-lapse still images. Right: kymographs of Mms6-GFP signals in maximum projection. **e** A FRAP experiment time course with a WT AMB-1 cell expressing MamK-GFP. Yellow brackets indicate the portion of the MamK-GFP filament designated for photobleaching. **g** A FRAP experiment time course with an ΔMIS cell expressing MamK-GFP where the bleached area moved from its original position

toward the cell pole. Yellow and blue brackets indicate the portion of the MamK-GFP filament designated for photobleaching. Blue brackets indicate the original bleaching area, and yellow brackets track the movement of the bleached area. The MamK-GFP is shown in false-color white in **e** and **g**. Scale bars = 1 μm in **e** and **g**. **f, h** Normalized (average mean and standard error of mean [SEM]) percent recovery of each strain's recovering cells with non-moving (**f**) and moving (**h**) bleached area. *n* = 23 (WT), 20 (ΔMIS), 28 (Δ*macA*), and 24 (Δ*mcaB*) cells in **f**. *n* = 15 (ΔMIS), 8 (Δ*macA*), and 10 (Δ*mcaB*) cells in **h**. See more details in Supplementary Fig. 14. The 50% mark is noted with a dashed orange line. The source data are provided as a Source Data file.

McaAB system also contributes to the differences in magnetosome dynamic positioning between AMB-1 and MSR-1 during the entire cell cycle.

Based on our study, we propose a possible model for McaAB-mediated dynamic positioning of magnetosomes (Fig. 9c). McaA localizes to the positive inner curvature of the cell as a dashed line via its N-terminal periplasmic VWA domain. McaA serves as a landmark to regulate the placement and distance of McaB-marked CMs through its cytoplasmic C-terminal domain. As a consequence, neighboring CMs are separated from each other allowing newly made EMs to be added at multiple locations of the chain. Without McaAB, the CMs are located closely together such that newly made EMs can only be added at the ends of the magnetosome chain. Furthermore, cryo-ET shows that

MamK cytoskeleton is composed of short filaments and located along the magnetosome chain in both WT and ΔMIS cells (Fig. 1c, d). Based on FRAP experiments, MamK filaments in WT display local recovery that can be caused by monomer turnover (depolymerization/polymerization), filament sliding, or formation of new filaments. In many McaAB deficient cells, MamK filaments recover and at the same time move across the cell, which might help position the magnetosomes in the midcell (Fig. 9c). We propose that the McaAB system localizes the turnover of MamK filaments to allow for new magnetosome addition in between pre-existing magnetosomes in WT AMB-1 (Fig. 9c).

Beyond elucidating an unknown aspect of magnetosome chain formation, our findings raise new questions regarding the cell biology of organelle formation and maintenance in bacteria. For instance, how

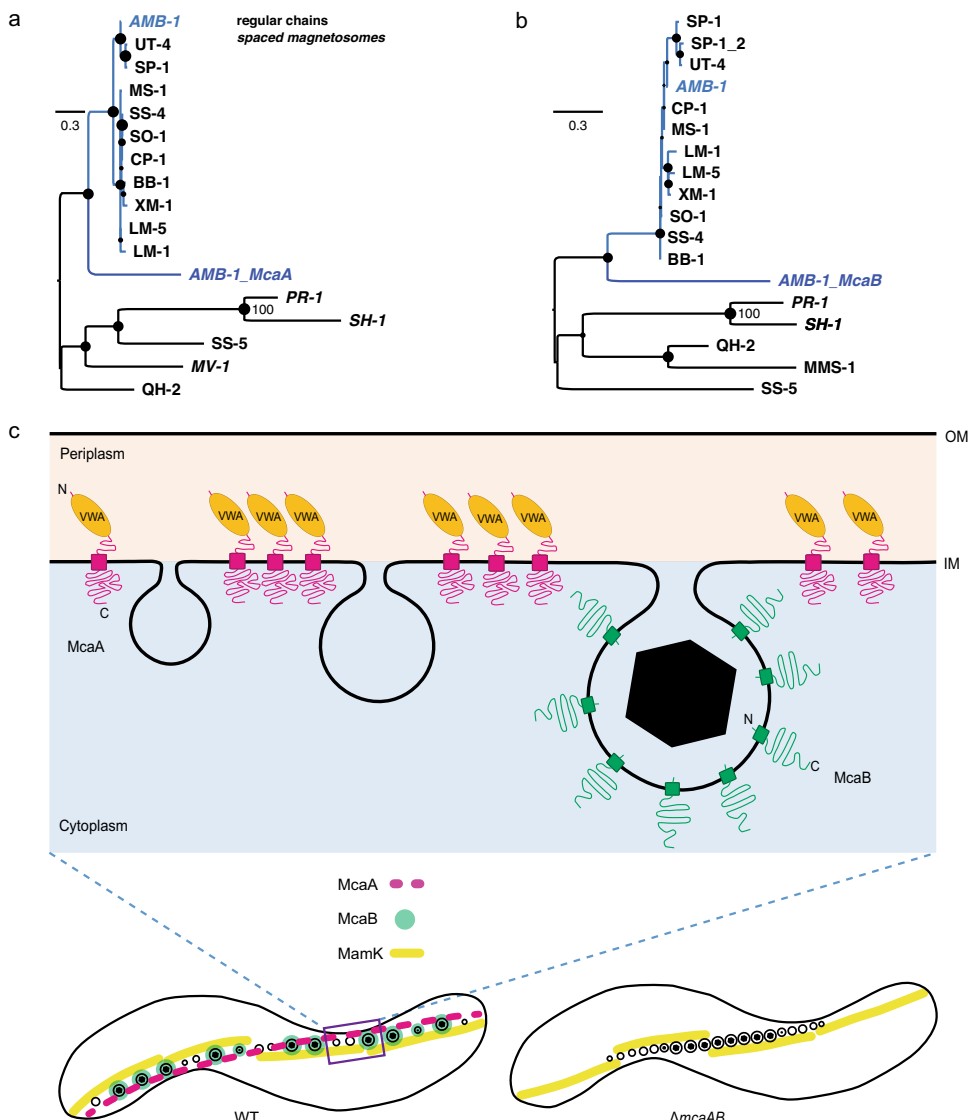

**Fig. 9 | Phylogenetic analysis and model of McaAB-mediated magnetosome chain assembly. a**, **b** Maximum likelihood trees showing the ancestry of McaA (**a**) and McaB (**b**) proteins in relation to their homologs in freshwater magnetotactic *Rhodospirillaceae* (blue clade) and the external groups of other *Proteobacteria*. All strains' accession numbers are given in Supplementary Dataset 1. Trees were drawn to scale and branch length refers to the number of substitutions per site. Robustness of the internal branches is symbolized by a circle whose size is proportional to the bootstrap value estimated from 500 nonparametric replicates. The magnetosome chains in these strains were previously characterized. If magnetosomes are spaced from each other similarly to strain AMB-1, names are in italics. **c** McaA serves as a landmark on the positively curved inner membrane and coordinates with McaB to control the location of CMs to the gap region of dashed McaA. As a consequence, the neighboring CMs are separated from each other, which allows the addition of newly formed EMs to multiple sites of the magnetosome chain in WT AMB-1. Alternatively, the CMs are located closely together without McaAB, leaving no space for the addition of newly formed EMs between CMs but at both ends of the magnetosome chain. McaAB also influences the dynamics of MamK filaments to control the dynamic positioning of magnetosomes during the whole cell cycle. OM outer membrane, IM inner membrane.

McaA detects the positive curvature of cytoplasmic membrane and localizes it as a dashed pattern remains a mystery. The unique localization pattern of McaA might be specific to helical-shaped bacterial cells (Supplementary Note 8). We confirmed that the localization of McaA is dependent on its VWA domain. Eukaryotic VWA-containing proteins are involved in a wide range of cellular functions, but they share the common feature of being involved in protein–protein interactions, many of which depend on divalent cations coordinated by the MIDAS motif[35]. Consistently, the MIDAS motif is essential for the location and function of McaA. VWA-domain proteins have been identified in some bacteria and archaea with different functions, but they are not well characterized[39–41]. These data suggest that there might be unknown proteins that partner with McaA to determine its specific localization.

In addition to coordinating with McaB to control the fragmented crystal chain assembly, McaA also helps to prevent magnetosome chain aggregation when *mamJ* and its homologs are deleted. It also assists in keeping magnetosomes to the positively curved membranes when *mamY* is deleted, indicating that McaA contributes to multiple aspects of magnetosome chain organization. While our genetic and cell biological studies support the links between McaA/B and MamK, as well as the connections between McaA and MamJ/Y, the exact interactions between them remain mysterious. We did not observe any direct interactions between McaA/B and MamK/J/Y through BACTH assays, which might be due to the non-detectable weak interactions or potential unknown intermediate proteins participating in this process. MamJ has been shown to directly interact with MamK through BACTH analysis in MSR-1[23] and it regulates the in vivo dynamics of MamK

filaments in AMB-1[24]. Recently, a curvature-inducing protein CcfM has been identified and characterized in MSR-1[42]. CcfM localizes in a filamentous pattern along the positively curved inner membrane by its coiled-coil motifs, and it also functions as an intermediate protein that links the interaction between MamY and MamK[42]. Whether CcfM plays any role in McaA localization and its interactions with MamK/J/Y still need to be investigated.

Although many MTB species from different taxa contain the McaA and McaB homologs, only AMB-1 and two other species of *Proteobacteria* form a fragmented magnetosome chain. Given our data, the evolutionary history of the fragmented chain formation strategy in AMB-1 may be explained by several scenarios. In the first one, the MIS could have been acquired after AMB-1 emergence from an unknown magnetotactic *Rhodospirillaceae* donor through a bacteriophage-mediated lateral gene transfer. Indeed, the MIS is flanked with putative phage-related proteins that appear to be very well conserved in *Rhodospirillaceae* spp. Even if this scenario is parsimonious because it minimizes the number of evolutionary events, the presence of homologs of several magnetosome genes in the MIS with partial synteny conservation and many transposases could also be evidence for an old duplication event. In this case, the MIS would be a partial remnant of one of the duplicated versions. The duplication event at the origin would be even more ancestral to the one that led to the emergence of the *lim* cluster[38]. However, given the known *Magnetospirillum* evolutionary history[38], this scenario would imply that many independent losses occurred over *Magnetospirillum* diversification. Assuming the latter scenario, fragmented chain formation would either be the trace of an ancestral strategy progressively replaced in the majority of lineages, or a recent one that emerged in AMB-1.

Our discoveries also highlight that previously undiscovered genes (*mca* and *mca-like* homologs) outside of MAI and conserved in diverse MTB species can play essential roles in magnetosome biosynthesis. The function of Mca-like proteins conserved in MTB remains to be elucidated; their proximity to the MAI, the conservation of their synteny, and the presence of the VWA domain in McaA-like proteins indicate that they probably play important role in magnetosome positioning along the magnetoskeleton.

It is notable that the action of two proteins is sufficient to fundamentally alter the assembly and organization of magnetosome chains in AMB-1 as compared to MSR-1, one of its closest relatives. We propose that the alternative mode of chain organization in AMB-1 may provide advantages that have led to its selective maintenance. Since magnetic particles are arranged as sub-chains along the length of AMB-1, daughter cells are ensured to inherit equal numbers of magnetic particles that are centrally positioned. In addition, the distribution and spacing of CMs and EMs may reduce the forces needed to separate magnetic particles. In contrast, MTB such as MSR-1, need to break the closely located continuous crystal chain in the middle and dynamically reposition the entire chain after cell division, which could be more energy-demanding than the stationary ones in WT AMB-1. However, as seen in our Cmag data, AMB-1 cells, as a population, align better in magnetic fields in the absence of *mcaAB*. Given the specific biological interventions required for their assembly, preserved magnetite or greigite chains are also considered an important criterion for magnetofossil recognition and characterization[13,14]. Thus, understanding the selective pressures that dictate the species-specific mechanisms of chain organization in modern-day organisms can provide much-needed insights into the conditional functions of magnetosomes across evolutionary time.

Finally, the McaAB system appears to functionally resemble the McdAB system that determines the positioning of carboxysomes[6], the $CO_2$-fixing protein organelles in cyanobacteria and proteobacteria. Both systems help to spatially position organelles along the longitudinal axis of the bacterial cell and ensure equal segregation of organelles into daughters during cell division. Importantly, both

systems contain two main proteins that organize themselves to coordinate organelle positioning. One protein (McdA/McaA) appears to function as an anchor to drag the organelle into position with the aid of a second protein (McdB/McaB) that is present on the organelle. However, the underlying mechanisms in the two systems are different. There is no homology between the proteins of these two systems. McdB localizes to carboxysomes, drives emergent oscillatory patterning of McdA on the nucleoid through directly binding to McdA, and stimulates McdA ATPase activity and its release from DNA. McaB localizes to CMs, but seems to not directly interact with McaA, which does not contain ATPase domain or DNA binding motif. In addition, the localization or function of McaA and McaB are not influenced by each other. Together, our findings add to the understanding of bacterial organelle positioning in general and highlight the complexity, diversity, and evolution of bacterial organelles.

## Methods

### Bacterial growth
The bacterial strains used in this study are described in Supplementary Table 4. The stock cultures of AMB-1 strains were prepared by picking single colonies into 1.5-mL of *Magnetospirillum* growth (MG) medium supplemented with 1/100 vol of Wolfe's vitamin solution and 30 μM ferric malate in 1.5-mL Eppendorf tubes and kept at 30 °C for 3 days[43]. The stock cultures were then used for larger volume growth with a dilution of 1:100. For Cmag measurements, TEM, cryo-ET, and fluorescent microscopy, 100 μL of stock cultures were added into 10 mL of MG medium in the 24-mL green-capped tubes and kept in a microaerobic glovebox (10% oxygen) at 30 °C for 1–2 days. For anaerobic growth conditions, 100 μL of stock cultures were added into 10 mL of MG medium in sealed Balch tubes. The MG medium was flushed with $N_2$ gas for 10 min and then autoclaved. The growth curves were measured with cells grown under anaerobic conditions. Doubling times were calculated from the growth curves of raw values for each strain shown in Supplementary Fig. 1a. For both microaerobic and anaerobic growth conditions, 1/100 vol of Wolfe's vitamin solution and 30 μM ferric malate were added to the MG medium just before inoculation with bacteria. For low iron growth condition, 100 μL of stock cultures were added in the 10-mL MG medium supplied with only 1/100 vol of Wolfe's vitamin solution, but no ferric malate, in the green-capped tubes that were treated with 0.375% oxalic acid (to remove the trace iron on the wall of glass tubes).

*Escherichia coli* strains DH5a, XL1-Blue, DHM1, and WM3064 were grown in Lysogeny broth (LB) medium with appropriate antibiotics. For *E. coli* strain WM3064, 300-μM diaminopimelic acid was added to the LB medium before inoculation with bacteria.

### Genetic manipulation
The genome sequence of *Magnetospirillum magneticum* AMB-1 (GenBank accession number NC_007626.1) was used for oligonucleotide design. Oligonucleotides were purchased from Elim Biopharm or Integrated DNA technologies. Plasmids were constructed by PCR amplifying DNA fragments of interest with the Phusion High Fidelity DNA Polymerase (New England Biolabs) or CloneAmp HiFi PCR Premix (Takara). All constructs were confirmed by sequencing in UC Berkeley DNA Sequencing Facility. DNA sequences were analyzed with ApE software[44]. All plasmids were introduced into AMB-1 by conjugation. The plasmids and primers used in this study are described in Supplementary Tables 5–11. The details about plasmids generation are as follows.

All deletion plasmids (pAK1037, pAK1121, pAK1151, pAK1152, pAK1188, pAK1189, pAK1190, pAK1191, pAK1224, pAK1225, and pAK1277, see more details in Supplementary Table 5) were generated similarly. Briefly, an approximately 800–1000 bp region upstream and downstream of the deleted gene or genomic region were PCR amplified from the AMB-1 genomic DNA using primer pairs (A, B) and (C, D),

respectively (Supplementary Table 6). The two PCR fragments were cloned into the SpeI restriction site of the pAK31 suicide plasmid using Gibson assembly to generate the deletion plasmids.

To generate pAK1240-pAK1252 (Supplementary Table 7) for BACTH studies, *mcaA*, *mcaB*, and *mamY* were PCR amplified using the primers listed in Supplementary Table 8 and were cloned into pKT25 or pKNT25 (the N or the C-termini of the T25 fragment) or pUT18 or pUT18C (the N or the C-termini of the T18 fragment) vectors in frame with the T25 and T18 fragment open reading frames by Gibson assembly.

Plasmid pAK1036 was constructed to express the fusion protein MamI-Halo under Tac promoter. Firstly, *mmsF* and HL4-Linker were PCR amplified using the primers listed in Supplementary Table 9 and cloned into pAK979[32] (digested with EcoRI) by Gibson assembly to generate pAK1032. Then, HL4-linker-*halo* was PCR amplified from pAK1032 using the primers listed in Supplementary Table 9 and inserted into pAK976[32] (digested with EcoRI and SpeI) by Gibson assembly to generate pAK1034. Finally, the *mamI* gene was PCR amplified from the AMB-1 genomic DNA using the primers listed in Supplementary Table 9 and cloned into pAK1034 (digested with EcoRI) by Gibson assembly to generate pAK1036.

Plasmid pAK1101 was constructed to express the fusion protein Mms6-Halo under Tac promoter. The *mms6* gene was PCR amplified from the AMB-1 genomic DNA using the primers listed in Supplementary Table 9 and cloned into pAK1034 (digested with EcoRI) by Gibson assembly.

Plasmid pAK1102 was constructed to express the fusion protein Mms6-GFP under Tac promoter with kanamycin resistance. The *mms6* gene was PCR amplified from the AMB-1 genomic DNA using the primers listed in Supplementary Table 9 and cloned into pAK22[31] (double digested with EcoRI and BamHI) by Gibson assembly.

Plasmid pAK1195 was constructed to express the fusion protein Mms6-GFP under Tac promoter with gentamycin resistance. The vector was PCR amplified from pAK1102 (except the kanamycin gene) using the primers listed in Supplementary Table 9. The gentamycin gene was PCR amplified from the plasmid pJN105 using the primers listed in Supplementary Table 9. Then the two DNA fragments were assembled by Gibson cloning.

Plasmid pAK1199 was constructed to complement the whole iR2 region deletion mutant. The whole iR2 region was PCR amplified from the AMB-1 genomic DNA using the primers listed in Supplementary Table 9, and cloned into pAK22 (double digested with EcoRI and SpeI) by Gibson assembly.

To construct pAK1200 (*gfp-mcaA*) and pAK1237 (*gfp-mcaB*), *ambRS23835* (*mcaA*) and *ambRS2475O* (*mcaB*) were PCR amplified from the AMB-1 genomic DNA using the primers listed in Supplementary Table 9, and inserted into pAK532 (digested with BamHI and SpeI) using Gibson assembly. To construct pAK1201 (*mcaA-gfp*) and pAK1238 (*mcaB-gfp*), *mcaA* and *mcaB* were PCR amplified from the AMB-1 genomic DNA using the primers listed in Supplementary Table 9, and inserted into pAK22 (digested with EcoRI and BamHI) using Gibson assembly.

Plasmid pAK1255 was constructed to express the fusion proteins of McaB-GFP and McaA-Halo under Tac promoter in the same cell. The *mcaA* gene and its ribosome binding site (*rbs-mcaA*) were PCR amplified from the AMB-1 genomic DNA using the primers listed in Supplementary Table 9. The *halo* gene was PCR amplified from pAK1036 using the primers listed in Supplementary Table 9. Then both PCR fragments were inserted into pAK1238 (digested with SpeI) using Gibson assembly.

Plasmid pAK1256 was constructed to express the fusion proteins of McaB-GFP and McaA-Halo under the native promoter of the *mcaAB* operon in the same cell. *mcaB* with its native promoter region and *rbs-mcaA* were PCR amplified from the AMB-1 genomic DNA using the primers listed in Supplementary Table 9. The *gfp* was PCR amplified

from pAK22 using the primers listed in Supplementary Table 9. Then all three PCR fragments were inserted into pAK1036 (digested with BamHI and HindII) using Gibson assembly.

Plasmids pAK1257-pAK1263 were constructed to express the mutated McaA that fused with GFP under Tac promoter. For pAK1257 and pAK1263, the N- and C- terminally truncated *mcaA* were PCR amplified from the AMB-1 genomic DNA, and inserted into pAK22 (digested with BamHI and EcoRI) using Gibson assembly. For pAK1258, pAK1260, pAK1261, and pAK1262, the upstream and downstream deleted regions of *mcaA* were PCR amplified from the AMB-1 genomic DNA, and inserted into pAK22 (digested with BamHI and EcoRI) using Gibson assembly. For pAK1259, the three conserved amino acids of the *mcaA* MIDAS motif (Asp-Xaa-Ser-Xaa-Ser (DXSXS), where X is any amino acid) were mutated into alanines, and the mutations were included in the primers. Then the fragments were PCR amplified from the AMB-1 genomic DNA and inserted into pAK22 (digested with BamHI and EcoRI) using Gibson assembly. All primers used to generate *mcaA* mutants were listed in Supplementary Table 10.

Plasmid pAK1270 was constructed to express the fusion proteins of McaB-GFP and Mms6-Halo under Tac promoter in the same cell. *rbs-mms6-halo* was PCR amplified from pAK1101 using the primers listed in Supplementary Table 9, and inserted into pAK1255 (digested with SpeI) using Gibson assembly.

## Deletion mutagenesis

A two-step homologous recombination method was used to generate deletion mutants in AMB-1 strains[43]. Briefly, the deletion plasmid was conjugated into AMB-1 strain using *E. coli* WM3064 donor strain. Colonies that had successfully integrated the plasmid were selected on MG agar plates containing 15 μg mL$^{-1}$ kanamycin. To select for colonies that had undergone a second recombination event to lose the integrated plasmid, a counter-selectable marker sacB, which is toxic in the presence of sucrose, was used. Colonies were then passed in 10 mL of growth media without kanamycin and plated on MG agar plates containing 2% sucrose. The resulting sucrose-resistant colonies were checked for successful deletions at their native locus by colony PCR with primers listed in Supplementary Table 11.

The whole MIS region was deleted in AMB-1 WT, ΔMAI, and Δ*mamJ*Δ*limJ* genetic backgrounds to generate the strains of ΔMIS, ΔMAIΔMIS, and Δ*mamJ*Δ*limJ*ΔMIS, respectively. The *mamJ-like* gene was deleted in WT and Δ*mamJ*Δ*limJ* genetic backgrounds to generate the strains of Δ*mamJ-like* and Δ*mamJ*Δ*limJ*Δ*mamJ-like*, respectively. The LD1 and LD2 regions of MIS were deleted in WT to create strains of ΔMIS_LD1 and ΔMIS_LD2, respectively. The iR1, iR2, iR3, and iR4 regions of MIS were deleted in WT to create strains of ΔiR1, ΔiR2, ΔiR3, and ΔiR4, respectively. The iR1, iR2, iR3, and iR4 regions of MIS were deleted in the Δ*mamJ*Δ*limJ* strain to create the triple deletions of Δ*mamJ*Δ*limJ*ΔiR1, Δ*mamJ*Δ*limJ*ΔiR2, Δ*mamJ*Δ*limJ*ΔiR3, Δ*mamJ*Δ*limJ*ΔiR4, respectively. The *mcaA* gene was in-frame deleted in WT and Δ*mamJ*Δ*limJ* genetic backgrounds to generate the strains of Δ*mcaA* and Δ*mamJ*Δ*limJ*Δ*mcaA*, respectively. The *mcaB* gene was in-frame deleted in WT and Δ*mamJ*Δ*limJ* genetic backgrounds to generate the strains of Δ*mcaB* and Δ*mamJ*Δ*limJ*Δ*mcaB*, respectively. The *mamY* gene was in-frame deleted in WT, ΔMIS, ΔiR2, Δ*mcaA*, and Δ*mcaB* genetic backgrounds to create the strains of Δ*mamY*, Δ*mamY*ΔMIS, Δ*mamY*ΔiR2, Δ*mamY*Δ*mcaA*, and Δ*mamY*Δ*mcaB*, respectively. The coordinates and magnetic phenotypes of these deletion mutants are listed in Supplementary Table 12.

## Cellular magnetic response

The optical density at 400 nm (OD$_{400}$) of AMB-1 cultures in the green-caped tubes was measured at 24 and 48 h using a spectrophotometer. A large magnet bar was placed parallel or perpendicular to the sample holder outside the spectrophotometer,

and the maximum and minimum $OD_{400}$ were recorded. The ratio of the maximum to the minimum was designated as AMB-1 cells' Cmag.

## Transmission electron microscopy (TEM)

For imaging the whole AMB-1 cells by TEM, 1-mL AMB-1 cells were taken from the 10-mL cultures that grew under different conditions. The 1-mL cells were pelleted and resuspended into 5–10 μL of MG medium. The resuspended cells were applied on a 400-mesh copper grid coated with Formvar and carbon films (Electron Microscopy Sciences). The grids were glow-discharged just before use. Then the air-dried cells were imaged on an FEI Tecnai 12 transmission electron microscope equipped with a 2k × 2k charge-coupled device camera (The Model 994 UltraScan®1000XP) at an accelerating voltage of 120 kV using Gatan Digital Micrograph. For crystal size quantification, individual crystals were measured by hand using ImageJ software. The longest axis between two parallel crystal faces is reported as crystal size or the length, and the axis perpendicular to that was considered the width. The shape factor represents the ratio of the short axis over the long axis.

## Statistics and reproducibility

All datasets were first analyzed for normality using the Shapiro–Wilk test (Supplementary Table 13). If the $P$ value is smaller than 0.05, the data tested do not form a normally distributed population. For checking the significant difference between two groups of samples the following criteria were used: If the datasets are normally distributed, two-sided unpaired Student $t$-tests were performed; if one or both datasets are randomly distributed, the two-tailed unpaired Mann–Whitney $U$ test was used (see the details including $P$ values in Supplementary Table 14). The statistical analyses were performed using GraphPad PRISM software.

All TEM micrographs, cryo-electron tomograms, and SIM images are representatives of the strain grown under the stated conditions. The similar magnetosome chain organization phenotypes on TEM micrographs and cryo-electron tomograms, as well as the similar protein localization phenotypes on SIM images, in WT and mutated AMB-1 cells were observed in two or more independent experiments. Similar localization of McaA, McaB, and Mms6 in different cellular fractions (e.g., Fig. 5d) was observed in three independent experiments.

## Cryo-electron tomography (cryo-ET)

For cryo-ET sample preparation, 3 mL of log-phase AMB-1 cultures were pelleted and resuspended in 30 μL of MG medium. The resuspended cells were mixed with 30 μL of 2× bovine serum albumin–treated 10-nm-diameter colloidal gold fiducial markers[45]. Then, 3 μL of this mixture was applied to a glow-discharged, 200 mesh holey carbon-coated copper Quantifoil grid (Ted Pella, Inc.) in a Vitrobot (FEI Company)[46]. The Vitrobot chamber was maintained at a temperature of 22 °C and humidity of 100%. Excess liquid was blotted off the grid with a blot force of 3, blot time of 2.5 s, and drain time of 1 s. The grid was then plunge-frozen in liquid ethane that was cooled with liquid nitrogen, and imaged by cryo-ET. The two-dimensional images of WT and ΔMIS cells were recorded using JEOL JEM–3100 FFC FEG TEM (JEOL Ltd.) equipped with a field emission gun electron source operating at 300 kV, an Omega energy filter (JEOL), and a K2 Summit counting electron detector camera (Gatan). Single-axis tilt series were collected using SerialEM software[47] from −60° to +60° with 1.5° increments, at a final magnification of 6000× corresponding to a pixel size of 0.56 nm at the specimen, and a defocus set to −15 μm under low dose conditions (a cumulative electron dose of ~120 e A$^{-2}$). Tomogram reconstructions were visualized using the IMOD software package[48]. Amira was used for the 3D model segmentation (Thermo Fisher Scientific).

## Size and location analysis of magnetosome membranes

The reconstructed tomograms were visualized using a 3dmod software package[48]. To evaluate the relative size and location of magnetosome membranes, the diameter of each magnetosome membrane was first measured. Briefly, The largest membrane section of individual magnetosomes was determined by walking through the tomographic slices. Three independent diameters were measured on the largest membrane sections, and the averaged values were considered as the size of the magnetosome membranes. Then the size of magnetosome membranes in each cell was sorted from largest to smallest and was nominated from 1 (largest) to 0 (smallest) accordingly. The first magnetosome membrane on the left of the chain was numbered 0, the middle one was numbered 1, and the last magnetosome membrane on the right of the chain was numbered 2. The location of other magnetosome membranes was nominated accordingly.

For measuring the distance between individual magnetosome membranes, the shortest distances between neighboring magnetosome membranes were found by working through the tomographic slices and manually measured by 3dmod.

## Structured illumination fluorescent microscopy (SIM)

To stain the genomic DNA, AMB-1 cells growing in 10-mL MG medium of the green-capped tubes were collected by 16,800 × g for 3 min. The cell pellets were resuspended in a 1-mL fresh MG medium and stained with 1.4 μM 4′,6-diamidino-2-phenylindole (DAPI) in dark at room temperature for 15 min, and then washed three times with fresh MG medium. After washing, the pellet cells were resuspended with 30–50 μL of MG medium and were immediately imaged by Carl Zeiss Elyra PS.1 structured illumination microscopy with objective lens Plan-APOCHROMAT 100×/1.46. 3D-SIM z-series were acquired at a total thickness of 2 μm with 100 nm z-step spacing. DAPI, GFP, JF549, and JF646 were excited by 405, 488, 561, and 642 nm lasers, respectively, and fluorescence from each fluorophore was acquired through 420–480, 495–550, 570–620, and LP655 nm bandpass filters, respectively. Raw images were acquired and processed using ZEN software (Zeiss). The processed images were then visualized using Imaris (Bitplane).

MamI-GFP localization patterns in WT and ΔMIS cells were manually measured using the Fiji software package[49]. The magnetosome chain showed a linear line across the cellular axis, so the end-to-end distance of the GFP fluorescence line was considered as the length of the magnetosome chain. A line that parallels the magnetosome chain was drawn from cell pole to pole, and this line was considered the length of the whole AMB-1 cell.

## Pulse-chase analysis

To study the addition of newly made EM into magnetosome chains over time, we applied pulse-chase analysis using Halo-tagged magnetosome proteins as magnetosome markers. Halo ligands can irreversibly bind to Halo proteins. Under standard growth conditions, WT or mutated AMB-1 cells expressing Halo-tagged magnetosome proteins were grown to an early exponential phase ($OD_{400}$ is ~0.05). Cells were pelleted and resuspended with 60-μL MG medium and mixed with 120-μL 5 μM pulse Halo-ligand JF549, kept at 30 °C for 2.5 h in dark to make sure the Halo proteins were stained saturated with JF549. Then the extra JF549 ligand was washed away with MG medium (three times, 10 min for each time). keep a few microliters of JF549-stained cells for SIM microscopy imaging. Put the rest cells back in a green-capped tube containing 10-mL MG medium, Kept in the 10% oxygen glovebox at 30 °C for 4 h in dark to allow new protein and new magnetosome production. Cells were then pelleted and resuspended with 60-μL MG medium, and mixed with 120-μL 5 μM chase Halo-ligand JF646, kept at 30 °C for 1 h in dark. Then the extra JF646 ligand was washed away with MG medium (three times, 10 min for each time) and imaged immediately with a SIM microscope. For the control group, after one quick

wash with MG medium, pulse-stained cells were fixed with 4% paraformaldehyde for 1 h at room temperature to prevent the production of new Halo-tagged proteins, then washed with MG medium three times and kept for the chase staining. For transition experiments from iron starvation to standard iron growth conditions, AMB-1 cells were first grown with an MG medium without added iron and stained with JF549, then were inoculated to the MG medium with iron and incubated in the 10% oxygen glovebox for 4–5 h before JF646 staining.

## Quantitative colocalization analysis of fluorescent-labeled magnetosome proteins

After image collection from the SIM microscope, the subcellular distributions of GFP- and Halo-labeled proteins or different-colored ligands during pulse-chase experiments were quantitatively analyzed by Pearson's correlation coefficient and Manders' colocalization coefficients[50] using the ImageJ JaCoP plugin (Supplementary Table 1). To avoid background and noise signals, only the region of interest (magnetosome chain) was cropped from the original image and measured for colocalization analysis.

## Fluorescence recovery after photobleaching (FRAP)

Ten milliliters of AMB-1 cells in the early exponential phase ($OD_{400}$ is ~0.05) were pelleted and resuspended in ~20 μL of MG medium. Then, 3 μL of concentrated AMB-1 cells were applied in a glass-bottom dish (MatTek Corporation) and then covered with 2% solidified agarose for cell immobilization. The 2% agarose covers were formed by spotting 400-μL melted 2% agarose prepared in MG medium to the hole of the glass-bottom dish and solidifying for more than 30 min.

FRAP experiments were carried out on an inverted Carl Zeiss LSM880 FCS laser scanning confocal microscope with an objective lens Plan-Apochromat 100×/1.40 Oil DIC. MamK-GFP filaments were imaged using 488 nm excitation at 0.4% laser power and fluorescence was acquired through a 490–600 nm bandpass filter. A small area of the filaments was bleached using 488 nm laser light at 100% laser power for 5 iterations with intervals of 30 s and 30 cycles. For each strain, images were captured through the LSM880 Zen software (Zeiss) and analyzed using Fiji[49].

## Time-lapse imaging using HILO microscopy

For sample preparation, round coverslips (Matsunami, 25-mm diameter, 0.12–0.17 mm thick) were used as the imaging support. The coverslip was coated with poly-L-lysine and 500 μL of culture was added to an Attofluor cell chamber (Thermo Fisher Scientific). Then, a 5-mm-thick gellan gum pad (containing 0.55% gellan gum and 0.08 mM $MgCl_2$ in MG liquid medium) on the top of the coverslip to sandwich the cells against the bottom coverslip during time-lapse imaging. After removing excess culture by a pipette, the chamber was filled with fresh MG liquid medium, and the top of the chamber was covered with another coverslip to allow adequate microaerobic conditions to support the growth of AMB-1 cells. The sample was set up under about 10% oxygen atmosphere.

Bacteria were imaged using a total internal reflection fluorescence (TIRF) microscopy-based system with an inverted microscope (Nikon, Ti-E) equipped with a 100× CFI Apo TIRF objective (Nikon) and a 1.5× C-mount adapter (Nikon). The sample was incubated at 28 °C using an incubation system for a microscope (Tokai Hit) during imaging. A 488-nm laser (Sapphire; Coherent) was used to illuminate the sample at an inclined angle. The angle of the laser beam was adjusted manually to optimize the signal-to-noise ratio. Images were acquired using a high-sensitivity electron multiplying charged-coupled device camera (iXon3; Andor) with EM and preamplifier gains of 296 and 2.4×, respectively. The Z-position was adjusted to the best focus and was maintained by using a Perfect Focus System (Nikon) during time-lapse imaging. The exposure times for GFP and bright-field images were 500

and 30 ms, respectively, at 2- or 3-min intervals, and the samples were illuminated only during exposure.

Images were processed using NIS Elements AR (Nikon) and ImageJ software. The only alterations to the time-lapse images were contrast adjustments using the brightness/contrast "auto" command of ImageJ. The NIS Elements AR "rotate" command was used to rotate cells into a uniform vertical orientation for kymograph analysis. Kymographs were generated using the NIS Elements AR command "show slice view" in maximum projection.

## Protein secondary structure prediction methods

Membrane topology prediction method CCTOP (ref or http://cctop.enzim.ttk.mta.hu) was used with the TM Filter and Signal Prediction. Signalp 4.1 (http://www.cbs.dtu.dk/services/SignalP-4.1/) and Phobius (https://phobius.sbc.su.se) were used for signal peptide prediction. SMART (http://smart.embl-heidelberg.de) and InterProScan (https://www.ebi.ac.uk/interpro/search/sequence/) were used for protein domain prediction.

## Cellular fractionation

WT AMB-1 cells expressing pAK1255 were first grown in 50-mL MG medium in conical tubes with a 1:100 dilution from stock cultures at 30 °C for 2 days and then grown in 2-L MG medium in a microaerobic glovebox (10% oxygen) for 2 days. These 2-L cells were pelleted by centrifugation at 8000 × g for 15 min and kept at −80 °C freezer for future use. Cell pellets were thawed on ice and resuspended in 5-mL ice-cold 25 mM Tris buffer (pH 7.0). Pepstatin A and leupeptin were added to a final concentration of 1 μg mL$^{-1}$, and PMSF was added to a final concentration of 1 mM. The resuspension was passed through a French press two times at 1000 psi. From this step, all samples were kept on ice or at 4 °C. Then, 20 μg mL$^{-1}$ DNase I and 2 mM $MgCl_2$ were added to the homogenate and incubated at 4 °C for 30 min. To separate the magnetosome fraction, the cell lysates were passed through a magnetized MACS LS column (Miltenyi Biotec Inc.) that was surrounded by magnets. After washing the column three times with 25 mM Tris buffer (pH 7.0), the magnets were removed and the magnetosome fraction was eluted in 5 mL of 25 mM Tris buffer (pH 7.0). To separate soluble and insoluble non-magnetic fractions, the column flow-through was centrifuged at 160,000 × g for 2 h. The sedimented membrane fraction was resuspended with 100 mM Tris buffer (pH 7.0) and both fractions were centrifuged a second time at 160,000 × g for 2 h. The resulting supernatant contained the non-magnetic soluble fraction and the resuspended pellet contained the non-magnetic insoluble fraction.

Cellular fractions were analyzed by SDS-PAGE. In brief, different fractions were mixed with 2× Laemmli Sample Buffer (Bio-Rad) and heated for 15 min at 95 °C. McaA-Halo and McaB-GFP fusion proteins were resolved by Bio-Rad stain-free any KDs gels before transfer to nitrocellulose membrane by electroblotting. For detecting Mms6 proteins, samples were loaded on Bio-Rad 16.5% Mini-PROTEAN Tris-Tricine Precast gels. Immunological detection was performed with primary antibodies, including anti-GFP polyclonal antibodies (1:2500 dilution, Abcam), anti-Mms6 polyclonal antibodies (1:2500 dilution, Produced by ProSci Inc), anti-HaloTag monoclonal antibody (1:1000 dilution, Promega), and secondary antibodies including F(ab')2-goat anti-mouse IgG (H+L) HRP-conjugate (1:5000 dilution, Invitrogen) and goat anti-rabbit IgG (H+L)-HRP-conjugate (1:10,000 dilution, Bio-Rad). The images were then captured and analyzed using Image Lab (Bio-Rad).

## Bacterial adenylate cyclase two-hybrid assay (BACTH)

The assay was performed as described in the Euromedex BACTH system kit manual. N- and C-terminal T18 and T25 fusions of McaA, McaB, or MamY proteins were constructed using plasmid pKT25, pKNT25, pUT18C, and pUT18 in E. coli K12 recA strain XL1-Blue. T18

and T18/T25 fusions with MamK or MamJ were generated previously[51]. Sequence-verified constructs expressing T18/T25 magnetosome protein fusions were co-transformed into competent *E. coli* DHM1 cells (lacking endogenous adenylate cyclase activity) in all pairwise combinations[52], then plated on LB agar plates containing 100 µg mL$^{-1}$ carbenicillin and 50 µg mL$^{-1}$ kanamycin, and incubated at 30 °C overnight. Several colonies of T18/T25 cotransformants were isolated and grown in LB liquid medium with 100 µg mL$^{-1}$ carbenicillin and 50 µg mL$^{-1}$ kanamycin overnight at 30 °C with 220 rpm shaking. Overnight cultures were spotted on indicator LB agar plates supplemented with 40 µg mL$^{-1}$ X-gal, 100 µg mL$^{-1}$ carbenicillin, 25 µg mL$^{-1}$ kanamycin, and 0.5 mM IPTG. Plates were incubated 24–48 h at 30 °C before imaging. Bacteria expressing interacting hybrid proteins will show blue, while bacteria expressing non-interacting proteins will remain white.

### Comparative genomics and molecular phylogenetics

McaA and McaB homologs were searched in other bacterial genomes available in public databases. Protein sequences were aligned against reference proteins and non-redundant protein sequences of the *refseq_protein* and *nr* NCBI databases respectively in October 2021 using the BLASTP algorithm, a word size of 6, and default scoring parameters. A similar task was performed using public genomic assemblies of MTB annotated with the Microscope platform[53]. BLAST hits with an expectation value below $5 \times 10^{-2}$ were further analyzed. First, pairwise sequence comparisons were performed using BLASTP (BLAST+ version 2.10.0). Sequence clustering was then performed with the Mmseqs2[54] clustering algorithm version 13.45111 to define groups of distant homologs using the default parameters, a sequence identity threshold of 30% and an alignment coverage of 80% for the longer sequence and for the shorter sequence.

A first phylogenetic tree was built to determine the monophyly of the different clusters and evaluate their taxonomic and phenotypic composition. For this task, the total 38 McaA and 31 McaB homologous sequences retrieved were aligned using MAFFT version 7.487[55]. Relaxed trimming on the alignments was then performed using BMGE[56], selecting the BLOSUM30 substitution matrix, a minimum block size of 2, and removing characters with more than 50% gaps. Maximum likelihood trees were built using IQ-TREE[57] version 2.1.3, the substitution model for each protein was selected by ModelFinder[58] with the Bayesian Information Criterion. The statistical support of the branches was estimated by a standard nonparametric bootstrapping approach implemented in IQ-TREE applying 500 replicates. All sequences retrieved belong to MTB from *Proteobacteria* classes and *Nitrospirae*. Non-Alphaproteobacteria members were used to form external groups and infer the ancestry of McaA and McaB genes compared to other clusters.

Guided by these first phylogenies, the second set of trees was built following the same approach using genome sequences of strains for which TEM images of the magnetosome chains are available and for which organization could be compared with that of *Magnetospirillum magneticum* AMB-1. Relationships between sequences and chain features were then inferred after collecting all metadata including TEM images published previously (Supplementary Table 1). The synteny analyses were further explored using the tools implemented in the Microscope[53] platform.

### Reporting summary

Further information on research design is available in the Nature Research Reporting Summary linked to this article.

## Data availability

All data are available within the article and supplementary files. Source Data are provided with this paper.

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

## Acknowledgements

We thank members of the Komeili lab for helpful discussions and suggestions. We thank Pranami Goswami for helping generate the plasmid pAK1037, Julia Borden for helping generate the plasmids pAK1101 and pAK1102, Pedro Leão for helping generate the plasmid pAK1121. We thank Elizabeth Montabana and Kenneth H. Downing for assistance with cryo-ET data collection using the cryo-EM facility at Lawrence Berkeley National Laboratory. We also thank Danielle Jorgens, Reena Zalpuri, and Guangwei Min from the UC Berkeley Electron Microscope Laboratory for assistance in electron microscopy data collection. We thank Steven

Ruzin and Denise Schichnes from the CNR Biological Imaging Facility at UC Berkeley for their technical support with fluorescent microscopes. The Zeiss Elyra PS.1 Super-Resolution microscope for SIM was supported in part by the National Institutes of Health S10 program under award number 1S10OD018136-01. We thank the INRA MIGALE bioinformatics platform (http://migale.jouy.inra.fr) for providing computational resources. We thank Michelle C. Chang from the Department of Chemistry at UC Berkeley for the kind gift of anti-Mms6 antibodies. Finally, we thank Luke D. Lavis from Howard Hughes Medical Institute Janelia Research Campus for providing free JF549 and JF646 Halo ligands before they are commercialized. A.K. was supported by an NIGMS Maximizing Investigators' Research Award (R35GM127114). M.A. was supported by a grant through the Fondation pour la Recherche Médicale (ARF201909009123). E.T.-C. (while a graduate student in Komeili lab) was supported by NSF Graduate Research Fellowship Program (DGE 110640). A.T. supported by JSPS KAKENHI (19H02868 and 17KK0145). C.L.M. and C.T.L. were supported by the French National Research Agency (ANCESMAG: ANR-20-CE92-0050). G.E. was supported by a SPUR grant from the Rausser College of Natural Resources at the University of California, Berkeley.

## Author contributions

J.W. conceived, performed, and analyzed most of the experiments. A.T. conducted the live-cell imaging analysis with HILO microscopy. C.L.M. and C.T.L. performed the phylogenetic analysis of McaA and McaB. G.E. designed and generated the small region (iR1-iR4) deletion plasmids, generated the deletion mutants of ΔiR1-ΔiR4 and Δ*mamJ*Δ*limJ*ΔiR1-Δ*mamJ*Δ*limJ*ΔiR4, and collected the TEM images of these mutants. K.P. generated the deletion mutants of ΔMIS and Δ*mamJ*Δ*limJ*ΔMIS, and collected the TEM images of these two mutants. M.A. discovered the crystal shape difference between WT and ΔMIS. E.T.-C. helped to establish the Halo-based pulse-chase experiment system. A.K. supervised the experimental design, data analysis, and data presentation. J.W., A.T., C.L.M., C.T.L., and A.K. wrote the manuscript, and all co-authors edited the manuscript.

## Competing interests

The authors declare no competing interests.
