## [Peer Review File · Nature Communications]

Reviewers' Comments:

Reviewer #1:

Remarks to the Author:

The manuscript "McaA and McaB control the dynamic positioning of a bacterial magnetic organelle" by Wan and coworker describes the identification and characterization of two novel proteins, McaA and McaB, in *Magnetospirillum magneticum* AMB-1. The strain AMB-1 is one of the model strains to study magnetosome formation and previous work has shown that a complex network of proteins, including cytoskeletal elements are required for magnetosome positioning and maturation. A surprising fact is that in closely related *Magnetospirillum gryphiswaldense* MSR-1 organization of the same organelles seems to be differently regulated. While in MSR-1 the magnetosome chain is central and continuous, in AMB-1 the magnetosomes are arranged in packages that are dispersed along the cell length. In this work the authors started by analyzing an extra genomic cluster (MIS) with so far uncharacterized genes. Deletion analysis revealed that the coefficient of magnetism (C_{mag}) was even higher in the deletion strain, explained by a continuous magnetosome chain much alike the situation in MSR-1. Thus, the differences in magnetosome formation between the strains may be caused by proteins encoded in MIS. Indeed, the authors show that deletion of two genes, designated McaA and McaB, are sufficient to cause the observed phenotype. Subcellular localization studies showed that McaA is positioned along the positive curvature of the plasma membrane and McaB localized to the magnetosome membrane. Together these proteins might create gaps for the magnetosomes and also influence the dynamics of the MamK filament.

This study contains many very carefully executed experiments that support largely the hypothesis. The microscopic experiments and the underlying genetics are excellent and the experiments shown to not contain conceptual problems. Thus, the manuscript is a very nice addition to the literature about magnetotactic bacteria and it helps to solve the mystery of the different organization and dynamics of magnetosomes in AMB-1 and MSR-1. I very much enjoyed reading this article. The work could still slightly improve by considering a few points:

Fig. 4: The fluorescent images are quite difficult to see and interpret at that size. McaA-GFP localization is difficult to see and judge. Ideally these images would be larger.

Line 223: Co-localization of two proteins (here McaB and Mms6) in such a tiny cell does not allow to judge about interaction or presence in the same compartment. Higher resolved images would be necessary. The fractionation assay is much better suited. However, there (Fig. 4 f) several bands are visible. This requires some explanation. What is the size of the fusion protein? Which is the full-length band in the blot and is the rest degradation?

The interaction of McaA with the positive curvature of the membrane is not sufficiently shown. Ideally this notion would be supported by additional experiments, including a 3D imaging approach.

The proposed interaction of McaA and/or McaB with MamK is still elusive. In a deletion background MamK shows altered filament dynamics, but why and how remains unclear.

The final model is therefore a good working hypothesis, but the data are not sufficient to support this. What about the interaction with MamK? The discussion could get in a bit more detail on the complex interaction network including MamK, MamY etc. functions.

Line 33: I am not convinced that "many" bacteria contain organelles. Indeed, one might argue that despite the membrane surrounded magnetosomes there are little examples of bacterial organelles (according to the classical definition of a membrane surrounded compartment).

Reviewer #2:

Remarks to the Author:

In this manuscript, the authors studied extensively the mechanisms that determine the chain organization of magnetosomes in the model magnetotactic bacteria *Magnetospirillum magneticum* AMB-1 and *Magnetospirillum gryphiswaldense* MSR-1. They reported two proteins, McaA and McaB, are responsible for the location and spacing of magnetosomes, especially newly added magnetosomes, as well as the magnetosomes dynamic positioning during the cell cycle. The manuscript was well presented in general and the study provides insight into magnetosome chain organization and the evolutionary history of magnetosome positioning systems in magnetotactic

bacteria. However, I have a few concerns about the data and interpretation, in addition to some minor comments.

Major comments:

Cryo-ET showed extra experimental evidence of magnetosome subcellular organization and interactions with filaments and membranes. But the presented Cryo-ET results were limited and were not interpreted in detail.

Bacterial adenylate cyclase two-hybrid (BACTH) assay results presented in this work were confusing and did not fully support microscopic findings. For example, McaA and B interactions and McaAB associations with MamK/J/Y proteins, which are the key observations of this work, were not fully verified by BACTH assays.

Growth assays or cell physiological characterization of WT and knockout mutants in this work should be presented.

The McaAB system appears to functionally resemble the McdAB system that determines the positioning of carboxysomes (MacCready et al. eLife 2018;7:e39723), the CO₂-fixing protein organelles in cyanobacterial and proteobacteria. Their similarities should be discussed in the manuscript in a whole context.

Other comments:

Line 106-107: it is important to show large views or multiple cells to indicate the representative location of crystals in WT and MIS-knockout.

Line 172, 184, 187, 213: specify which supplementary results.

Line 175: Does it mean there are random mutations in the lab strain? Any other missing genes and mutations identified in the lab strain genome? Does it affect cell phenotype?

Fig. 1c: Clarify the two-end arrows in the corresponding Results and figure legend.

Fig. 2d: The extra green lines in JF646 indicate membrane localization, apart from the chain organization. What is the reason? JF646 signal was not completely located at two ends. Was this conclusion supported by any statistical analysis?

Line 195: it is useful to conduct colocalization analysis of McaA and magnetosomes (fluorescently tagged Mms6 or MamI).

Line 229: McaB is not associated with magnetosome membranes. McaB associates with CMs specifically, not localize to EMs.

Line 241, Fig. 5a: it is better to use the specific McaA-KO and McaB-KO or McaAB-KO mutants that express MamI-GFP, instead of MIS-KO and iR2-KO. The same for FRAP experiments in Fig. 7g.

Supplementary Fig. 14: add mcaB complementation results.

Supplementary Fig. 12 was not described in order.

Fig. 6: specify AMB-1 and MSR-1 strains for the corresponding mutants.

Line 368: tune down the conclusive statement "uncovered the mechanisms of ..."

Line 382: the short filaments of MamK cytoskeleton shown in cryo-ET was not clearly described in Results.

Line 391: what is "cell biological mechanisms"?

Line 626: "McdA, McdB" should be "McaA, McaB".

Reviewer #3:

Remarks to the Author:

The authors conducted a very robust and thorough investigation of the functions of McaA and McaB in AMB-1. They found that the proteins function in the addition of new magnetosomes to a chain and determined a key step in the formation of the magnetosome chain in MTB. They compare the assembly and organization of magnetosome chain in AMB-1 to other MTB. The work is novel and will be of interest to the field of environmental microbiology, particularly those interested in MTB.

Their manuscript is detailed, thorough and well written (except for a few minor errors that can be easily corrected in edition), easy to read. References are appropriate for the research. Experimentation is well planned and conducted in an appropriate manner. Data supports conclusions. Figures and table are high-quality, easy to read and understand. Supplemental materials are extensive and appropriate for the research. Materials and Methods adequately describe the research conducted here including bacterial growth, genetic manipulation, microscopy, enzyme assays, and comparative genomics.

Point-by-point response to the reviewer's comments (manuscript NCOMMS-22-10247)

Reviewer #1 (Remarks to the Author):

The manuscript "McaA and McaB control the dynamic positioning of a bacterial magnetic organelle" by Wan and coworker describes the identification and characterization of two novel proteins, McaA and McaB, in *Magnetospirillum magneticum* AMB-1. The strain AMB-1 is one of the model strains to study magnetosome formation and previous work has shown that a complex network of proteins, including cytoskeletal elements are required for magnetosome positioning and maturation. A surprising fact is that in closely related *Magnetospirillum gryphiswaldense* MSR-1 organization of the same organelles seems to be differently regulated. While in MSR-1 the magnetosome chain is central and continuous, in AMB-1 the magnetosomes are arranged in packages that are dispersed along the cell length. In this work the authors started by analyzing an extra genomic cluster (MIS) with so far uncharacterized genes. Deletion analysis revealed that the coefficient of magnetism (Cmag) was even higher in the deletion strain, explained by a continuous magnetosome chain much alike the situation in MSR-1. Thus, the differences in magnetosome formation between the strains may be caused by proteins encoded in MIS. Indeed, the authors show that deletion of two genes, designated McaA and McaB, are sufficient to cause the observed phenotype. Subcellular localization studies showed that McaA is positioned along the positive curvature of the plasma membrane and McaB localized to the magnetosome membrane. Together these proteins might create gaps for the magnetosomes and also influence the dynamics of the MamK filament.

This study contains many very carefully executed experiments that support largely the hypothesis. The microscopic experiments and the underlying genetics are excellent and the experiments shown to not contain conceptual problems. Thus, the manuscript is a very nice addition to the literature about magnetotactic bacteria and it helps to solve the mystery of the different organization and dynamics of magnetosomes in AMB-1 and MSR-1. I very much enjoyed reading this article. The work could still slightly improve by considering a few points:

Thanks for your positive comments and constructive suggestions.

1) Fig. 4: The fluorescent images are quite difficult to see and interpret at that size. McaA-GFP localization is difficult to see and judge. Ideally these images would be larger.

This is a good point. As suggested, we now separated Fig. 4 into two figures (Fig. 4 and Fig. 5). The fluorescent images are larger and presented with separate channels. We also added single consecutive z-slices of the 3D-SIM images (Fig. 4c, 4h) and 3D movies (Supplementary movie 3, 4) that show the localization of McaA-GFP in WT and $\Delta MAI\Delta MIS$ cells.

2) Line 223: Co-localization of two proteins (here McaB and Mms6) in such a tiny cell does not allow to judge about interaction or presence in the same compartment. Higher resolved images would be necessary.

The reviewer is absolutely correct that the co-localization of two proteins does not allow us to judge interaction. And we did not make any conclusion about the interaction of McaB and Mms6.

The localization pattern of Mms6 has been addressed in previous work.

We now added “The association of Mms6 with CMs has been addressed previously in AMB-1 through localization analysis under different growth conditions and correlative fluorescent and TEM microscopy analysis³⁷” in lines 231-233.

Mms6 and McaB have similar localization patterns under different growth conditions. Both localize to the magnetosome chain as patches under standard growth conditions that produce both EMs and CMs, but do not localize to the magnetosome chain under low iron growth conditions that only produce EMs, indicating association with CMs and not EMs.

Importantly, structured illumination microscope (SIM) we used to image the AMB-1 cells in this study is a super-resolution microscopy technique with a resolution of about 120 nm. We believe the co-localization of signal between McaB and Mms6 provides another piece of evidence for the association of McaB with CMs as Mms6.

Other types of super-resolution optical microscopy techniques, such as photoactivated localization microscopy (PALM) or stochastic optical reconstruction microscopy (STORM), can reach up to a 20 nm resolution. But two-color PALM/STORM techniques are not widely available yet.

The fractionation assay is much better suited. However, there (Fig. 4 f) several bands are visible. This requires some explanation. What is the size of the fusion protein? Which is the full-length band in the blot and is the rest degradation?

The reviewer is correct that the fractionation assay is another important piece of evidence for the association of McaB with CMs. We have now added more explanation in the figure legends about the immunoblotting result as suggested. We also added control experiments in supplemental figure 10e-h.

Please see the legend of figure 5d: “McaA-Halo was detected with anti-Halo antibodies, McaB-GFP was detected with anti-GFP antibodies, and Mms6 was detected with anti-Mms6 antibodies. Full-length McaA-Halo (~118 kDa) and McaB-GFP (~52 kDa) proteins are marked with a circle. The unknown McaA-Halo related bands are marked with an arrow. The four proteolytically processed Mms6 fragments are marked with a right brace. See more details (uncropped blots) and controls in supplemental Figure 10e-h.”

3) The interaction of McaA with the positive curvature of the membrane is not sufficiently shown. Ideally this notion would be supported by additional experiments, including a 3D imaging approach.

This is a very important point. The SIM images generated in this manuscript are maximum-intensity projections of 3D Z-stacks. We have now specified it in the main text, figure legends, and methods.

Please see lines 143-144 in the main text: “Three-dimensional (3D) structured illumination fluorescent microscopy (SIM)”. And we have changed “SIM micrographs” into “Representative maximum-intensity projection of 3D-SIM micrographs” in the figure legends. Please see line 576 in the methods: “3D-SIM z-series were acquired at a total thickness of 2 μ m with 100 nm z-step spacing.”

As suggested, we also added more explanation and single consecutive z-slices of the SIM images to sufficiently show the localization pattern of McaA-GFP. Please see lines 197-199 “Specifically, McaA is closest to the regions of the cell envelope that are bent inward toward the cytoplasm, forming a line covering the shortest distance from cell pole to pole (Fig. 4c and Supplementary Movie 3).” Please also see Fig. 4h, and Supplementary Movie 4.

4) The proposed interaction of McaA and/or McaB with MamK is still elusive.

This is an excellent point and worthy of future studies. Our fluorescent microscopy and genetic studies indicate potential connections between McaA/B and MamK, but the bacterial two-hybrid assays show negative results for the direct interactions between them. We have briefly explained the reasons for the negative results in our BACTH assays. Please see lines 248 -251: “Bacterial adenylate cyclase two-hybrid (BACTH) assays did not show any positive interactions between McaA and McaB (Table 1 and Supplementary Fig. 11a), indicating that either the fusion proteins do not interact strongly, are nonfunctional in the context of BACTH, or unknown intermediate proteins are needed to facilitate their interactions.”

In a deletion background MamK shows altered filament dynamics, but why and how remains unclear.

The reviewer is absolutely correct and we considered it throughout our work. We observed that the dynamics of MamK filaments are very different in WT and the McaAB deletion mutants. For the reasons of such differences, we thought it could be related to the dynamic localization of magnetosomes in the cell. Please see lines 393-399 in the discussion: “Based on FRAP experiments, MamK filaments in WT display local recovery that can be caused by monomer turnover (depolymerization/polymerization), filament sliding, or formation of new filaments. In many McaAB deficient cells, MamK filaments recover and at the same time move across the cell, which might help position the magnetosomes in the midcell (Fig. 9c). We propose that the McaAB system localises the turnover of MamK filaments to allow for new magnetosome addition in between pre-existing magnetosomes in WT AMB-1 (Fig. 9c).”

As mentioned above, McaA and McaB might not directly interact with MamK. The potential unknown proteins that regulate the dynamics of MamK filaments still need to be discovered. How the dynamics of MamK filaments change will be addressed in our future studies.

5) The final model is therefore a good working hypothesis, but the data are not sufficient to support this. What about the interaction with MamK? The discussion could get in a bit more detail on the complex interaction network including MamK, MamY etc. functions.

This is just one possible model, We now have specified this point in the discussion. Please see line 385: “we propose a possible model for...”.

Whether there are direct interactions between McaA/B and MamK/J/Y is not clear. As suggested, we now added more explanations about the potential connections between McaA/B and MamK/J/Y in the discussion, Please see lines 415-421: “While our genetic and cell biological studies support the links between McaA/B and MamK, also McaA and MamJ/Y, the exact interactions between them remain mysterious. We did not observe any direct interactions between McaA/B and MamK/J/Y through BACTH assays, which might be due to the non-detectable weak interactions or potential unknown intermediate proteins participating in this process. MamJ has been shown to directly interact with MamK through BACTH analysis in MSR-1²³ and it regulates the *in vivo* dynamics of MamK filaments in AMB-1²⁴”

The interaction network between these proteins is more complicated than we thought, which needs to be addressed in our future studies.

6) Line 33: I am not convinced that “many” bacteria contain organelles. Indeed, one might argue that despite the membrane surrounded magnetosomes there are little examples of bacterial organelles (according to the classical definition of a membrane surrounded compartment).

The organelles we mentioned here actually include membrane-bound and membrane-less organelles. The referenced paper (Organelle Formation in Bacteria and Archaea) talked about some known bacteria

membrane-bound organelles, including magnetosomes, ferresomes, anammoxosomes, and some membrane-bound storage granules. We now added more referenced papers about some known membrane-less organelles (i.e. carboxysomes, PHB granules, phosphate granules) and many unknown bacteria “organelle-like” structures (Uncharacterized Bacterial Structures Revealed by Electron Cryotomography). Please see line 31.

Reviewer #2 (Remarks to the Author):

In this manuscript, the authors studied extensively the mechanisms that determine the chain organization of magnetosomes in the model magnetotactic bacteria *Magnetospirillum magneticum* AMB-1 and *Magnetospirillum gryphiswaldense* MSR-1. They reported two proteins, McaA and McaB, are responsible for the location and spacing of magnetosomes, especially newly added magnetosomes, as well as the magnetosomes dynamic positioning during the cell cycle. The manuscript was well presented in general and the study provides insight into magnetosome chain organization and the evolutionary history of magnetosome positioning systems in magnetotactic bacteria. However, I have a few concerns about the data and interpretation, in addition to some minor comments.

Thanks for your positive comments and constructive suggestions.

Major comments:

1) Cryo-ET showed extra experimental evidence of magnetosome subcellular organization and interactions with filaments and membranes. But the presented Cryo-ET results were limited and were not interpreted in detail.

The reviewer is correct that a limitation of cryo-ET is the amount of data collection. This technique is extremely time and resource intensive, only a small number of WT and mutant cells could be studied (12 Δ MIS cells and 17 WT AMB-1 cells in this study). However, as the reviewer mentioned cryo-ET is an extra experimental approach we used to examine WT and Δ MIS cells. Even though a limited number of cells were imaged by cryo-ET, our data should be representative of the total population, since the crystal production phenotype (i.e. size distribution of the crystals and position in the cell) from our cryo-ET results is consistent with the other techniques that sample with hundreds or more cells, including conventional TEM and Cmag measurement.

As suggested, we now added more interpretation of the cryo-ET data in detail. Please see lines 116-121: “Magnetosomes are arranged as a chain that is flanked by a network of short filaments in both strains. Specifically, magnetosomes are located to the positive inner curvature of the cell (displayed along the area that are bent inward toward the cytoplasmic membrane), MamK filaments run parallel to two to five individual magnetosomes along the chain and do not show an obvious spatial position pattern in both WT and Δ MIS strains (Fig. 1c, d and Supplementary Movie 1, 2).”

We also added 3D movies of the cryo-ET data. Please see Supplementary Movies 1 and 2.

2) Bacterial adenylate cyclase two-hybrid (BACTH) assay results presented in this work were confusing and did not fully support microscopic findings. For example, McaA and B interactions and McaAB

associations with MamK/J/Y proteins, which are the key observations of this work, were not fully verified by BACTH assays.

The reviewer is correct that our genetic and cell biological studies suggest potential connections between McaA/B and MamK/J/Y. But our bacterial two-hybrid assays show negative results for the direct interactions between them. We have explained the reasons in the results (lines 248-251): “Bacterial adenylate cyclase two-hybrid (BACTH) assays did not show any positive interactions between McaA and McaB (Table 1 and Supplementary Fig. 11a), indicating that either the fusion proteins do not interact strongly, are nonfunctional in the context of BACTH, or unknown intermediate proteins are needed to facilitate their interactions.”

We now added more explanation in the discussion Lines (415-421): “While our genetic and cell biological studies clearly support the links between McaA/B and MamK, also McaA and MamJ/Y, the exact interactions between them remain mysterious. We did not observe any direct interactions between McaA/B and MamK/J/Y through BACTH assays, which might be due to the non-detectable weak interactions or potential unknown intermediate proteins participating in this process.”

The potential intermediate proteins between them will be addressed in our future studies.

3) Growth assays or cell physiological characterization of WT and knockout mutants in this work should be presented.

This is a good point. The growth curves of WT and Δ MIS strains have now been added as supplementary figure 1a. We also added the description in the results and methods section.

Please see lines 96-97 in the results: “The growth curves of Wild-type (WT) and Δ MIS strains are similar (Supplementary Fig. 1a and Supplementary Note 1).”

Supplementary Note 1: “**1. Growth curve characterization of WT and Δ MIS cells**

After obtaining the MIS deletion mutant, we first characterized the growth curve of Wild-type (WT) and Δ MIS strains. The lag phases and stationary phases are similar for both strains, but the log phases are a little bit different (Supplementary Fig. 1a). While the doubling time of WT in the log phase is about 5.5h, Δ MIS is about 4.1h. One possible explanation for this difference could be because there is less DNA to replicate for the Δ MIS strain.”.

Please see lines 488-490 in the methods: “The growth curves were measured with cells grown under anaerobic conditions. Doubling times were calculated from the growth curves of raw values for each strain shown in Supplementary Fig. 1a.”.

4) The McaAB system appears to functionally resemble the McdAB system that determines the positioning of carboxysomes (MacCready et al. eLife 2018;7:e39723), the CO₂-fixing protein organelles in cyanobacterial and proteobacteria. Their similarities should be discussed in the manuscript in a whole context.

As suggested, we have added the similarities and differences between the McaAB and McdAB systems in the discussion. Please see lines 465-479: “Finally, the McaAB system appears to functionally resemble the McdAB system that determines the positioning of carboxysomes⁶, the CO₂-fixing protein organelles in cyanobacteria and proteobacteria. Both systems help to spatially position organelles along the longitudinal axis of the bacterial cell and ensure equal segregation of organelles into daughters during cell division. Importantly, both systems contain two main proteins that organize themselves to coordinate organelle positioning. One protein (McdA/McaA) appears to function as an anchor to drag

the organelle into position with the aid of a second protein (McdB/McaB) that is present on the organelle. However, the underlying mechanisms in the two systems are different. There is no homology between the proteins of these two systems. McdB localizes to carboxysomes, drives emergent oscillatory patterning of McdA on the nucleoid through directly binding to McdA, and stimulates McdA ATPase activity and its release from DNA. McaB localizes to CMs, but seems to not directly interact with McaA, which does not contain ATPase domain or DNA binding motif. Additionally, the localization or function of McaA and McaB are not influenced by each other. Together, our findings add to the understanding of bacterial organelle positioning in general and highlight the complexity, diversity, and evolution of bacterial organelles.”

Other comments:

1) Line 106-107: it is important to show large views or multiple cells to indicate the representative location of crystals in WT and MIS-knockout.

We have modified it as suggested. We included large views (including multiple cells) to show the location of crystals in WT and Δ MIS in Supplementary Fig. 1b and Supplementary Fig. 2b.

2) Line 172, 184, 187, 213: specify which supplementary results.

We have now specified the “supplementary results” as “Supplementary Notes 1-8”.

3) Line 175: Does it mean there are random mutations in the lab strain? Any other missing genes and mutations identified in the lab strain genome? Does it affect cell phenotype?

We did notice some differences between our lab AMB-1 strain and the NCBI reference genome based on our preliminary genomic sequencing results. But the genomic sequences of WT and Δ MIS AMB-1 strains in our lab are very similar. We also carefully performed complementation experiments for the mutants in this manuscript to confirm the phenotypes are due to the deleted region or genes.

4) Fig. 1c: Clarify the two-end arrows in the corresponding Results and figure legend.

As suggested, we added, “ (magenta two-end arrows in Fig. 1c, d)” in the results (line 166) and “(the magenta two-end arrows on the tomographic slices of c and d)” in the legend of Fig. 1c, d.

5) Fig. 2d: The extra green lines in JF646 indicate membrane localization, apart from the chain organization. What is the reason? JF646 signal was not completely located at two ends. Was this conclusion supported by any statistical analysis?

Mami is a membrane protein that is needed for empty magnetosome invagination from the cytoplasmic membrane. Mami has been shown only located to the cytoplasmic membrane in the mutant that does not produce magnetosomes, and is located to both the cytoplasmic membrane and the magnetosome chain in WT AMB-1 ¹ (Fig. 3 of the reference 1) ² (Fig. 5 of the reference 2) . Thus It is normal to observe Mami-GFP signals both in the cytoplasmic membrane and the magnetosome chain (Fig. 2b). We have now added “Additionally, Mami-GFP was also observed at the cytoplasmic membrane, outlining the cell with a weak fluorescence (Fig. 2b).” in the main text (line 147-149), and “Mami-GFP proteins are located in the magnetosome chain and cytoplasmic membrane. Here and below: DAPI (4',6-Diamidino-2-Phenylindole) is a fluorescent dye that binds to DNA, and used as an indicator of AMB-1 cell contour.” in the legend of Fig. 2b.

JF646 signals represent newly-made Maml proteins, and it localizes to both the cytoplasmic membrane and new magnetosomes. If we only consider the JF646 signals on the magnetosome chain, the JF646 signals were mainly flanking the continuous old magnetosome chain that was stained with JF549. This conclusion is supported by the statistical analysis (low colocalization coefficients between JF549 and JF646) in Supplementary Table 1.

6) Line 195: it is useful to conduct colocalization analysis of McaA and magnetosomes (fluorescently tagged Mms6 or Maml).

We observed that McaB colocalizes with Mms6, and McaB is located in the gaps of the dashed McaA, indicating Mms6-marked CMs might also be located in the gaps of the dashed McaA. Maml is localized to both the EMs and CMs, McaA might partially colocalize with Maml. The reviewer is correct that this colocalization analysis is useful but is not crucial for the conclusions of the relationship between McaA and magnetosomes.

7) Line 229: McaB is not associated with magnetosome membranes. McaB associates with CMs specifically, not localize to EMs.

As suggested, we now specified it in line 241: “McaB with **crystal-containing magnetosomes**”.

8) Line 241, Fig. 5a: it is better to use the specific McaA-KO and McaB-KO or McaAB-KO mutants that express Maml-GFP, instead of MIS-KO and iR2-KO. The same for FRAP experiments in Fig. 7g.

iR2-KO is actually McaAB-KO mutants. We have now specified it in the text and figures. Please see line 202: “ Δ iR2 (**also called Δ mcaAB**)”, Figure 4e, 5a, 6a, 6c, and Supplementary Figure 5b, 9b, 10b, 10c.

For the FRAP experiments, we examined MIS-KO, as well as McaA-KO and McaB-KO. Please see Fig 8F, h, and Supplementary Figure 14a.

9) Supplementary Fig. 14: add mcaB complementation results.

Δ mamJ Δ limJ Δ mcaB and Δ mamJ Δ limJ have similar magnetosome chain organization phenotypes, indicating McaB is not the genetic element that contributes to magnetosome chain organization in Δ mamJ Δ limJ. It is not necessary to perform a McaB complementation experiment here.

10) Supplementary Fig. 12 was not described in order.

We now changed Supplementary Fig. 12 into Supplementary Fig. 16 and added a short description in the discussion. Please see lines 403-404: “**The unique localization pattern of McaA might be specific to helical-shaped bacterial cells (Supplementary Note 8).**”

11) Fig. 6: specify AMB-1 and MSR-1 strains for the corresponding mutants.

We only used AMB-1 strain in this study, and we did not use MSR-1 strain at all.

12) Line 368: tune down the conclusive statement “uncovered the mechanisms of ...”

As suggested, we changed “uncovered the mechanisms of ...” into “**discovered ...**” in line 379.

13) Line 382: the short filaments of MamK cytoskeleton shown in cryo-ET was not clearly described in Results.

As suggested, we added more descriptions to the results. Please see lines 116-121: “Magnetosomes are arranged as a chain that is flanked by a network of short filaments in both strains. Specifically, magnetosomes are both located to the positive inner curvature of the cell (displayed along the area that are bent inward toward the cytoplasmic membrane), MamK filaments run parallel to two to five individual magnetosomes along the chain and do not show an obvious spatial position pattern in both WT and Δ MIS strains (Fig. 1c, d and Supplementary movie 1, 2).”

Please see the legend in Figures 1c and 1d: “Magenta arrows point to the MamK filaments on the tomographic slices.”

14) Line 391: what is “cell biological mechanisms”?

It means the mechanisms of the cell's biological processes that are created for developing, maintaining, and regulating the organelle's behavior and function. These include membrane remodeling, protein sorting, organelle positioning, and organelle segregation. To simplify it, we now changed it to “cell biology” in line 401.

15) Line 626: “McdA, McdB” should be “McaA, McaB”.

As suggested, we have corrected it.

Reviewer #3 (Remarks to the Author):

The authors conducted a very robust and thorough investigation of the functions of McaA and McaB in AMB-1. They found that the proteins function in the addition of new magnetosomes to a chain and determined a key step in the formation of the magnetosome chain in MTB. They compare the assembly and organization of magnetosome chain in AMB-1 to other MTB. The work is novel and will be of interest to the field of environmental microbiology, particularly those interested in MTB.

Their manuscript is detailed, thorough and well written (except for a few minor errors that can be easily corrected in edition), easy to read. References are appropriate for the research. Experimentation is well planned and conducted in an appropriate manner. Data supports conclusions. Figures and table are high-quality, easy to read and understand. Supplemental materials are extensive and appropriate for the research. Materials and Methods adequately describe the research conducted here including bacterial growth, genetic manipulation, microscopy, enzyme assays, and comparative genomics.

Thank you for your positive comments! We have tried to correct some minor errors. And we are happy to correct the minor errors if you have some specific ones.

1. Murat D, Quinlan A, Vali H, Komeili A. Comprehensive genetic dissection of the magnetosome gene island reveals the step-wise assembly of a prokaryotic organelle. *Proc Natl Acad Sci U S A* **107**, 5593-5598 (2010).
2. Quinlan A, Murat D, Vali H, Komeili A. The HtrA/DegP family protease MamE is a bifunctional protein with roles in magnetosome protein localization and magnetite biomineralization. *Mol Microbiol* **80**, 1075-1087 (2011).

Reviewers' Comments:

Reviewer #1:

Remarks to the Author:

The authors have thoroughly revised their work and satisfyingly answered all points raised during the review. I support publication of this manuscript. I just had problems understanding one sentence (see below). Maybe a brief revision could clarify this?

Congratulations to all authors for this nice piece of work.

Line 416/417 : "While our genetic and cell biological studies support the links between McaA/B and MamK, also McaA and MamJ/Y, the exact interactions between them remain mysterious."

I am not sure if I understand that sentence correctly. Does it mean interaction of McaA/B and MamK as well as interaction between McaA and MamJ/Y?

Reviewer #2:

Remarks to the Author:

My comments have been addressed appropriately and I have no further concerns.

Point-by-point response to the reviewer's comments (manuscript NCOMMS-22-10247A)

Reviewer #1 (Remarks to the Author):

The authors have thoroughly revised their work and satisfyingly answered all points raised during the review. I support publication of this manuscript. I just had problems understanding one sentence (see below). Maybe a brief revision could clarify this?
Congratulations to all authors for this nice piece of work.

Line 416/417 : "While our genetic and cell biological studies support the links between McaA/B and MamK, also McaA and MamJ/Y, the exact interactions between them remain mysterious."
I am not sure if I understand that sentence correctly. Does it mean interaction of McaA/B and MamK as well as interaction between McaA and MamJ/Y?

Thanks for your kind and positive comments. To clarify this, we now changed that sentence into "While our genetic and cell biological studies support the links between McaA/B and MamK, as well as the connections between McaA and MamJ/Y, the exact interactions between them remain mysterious" as suggested. Please see the line 415-418 in the discussion.

Reviewer #2 (Remarks to the Author):

My comments have been addressed appropriately and I have no further concerns.

Thank you for your positive comments!